# Identification of Proteins Differentially Expressed by Adipose-derived Mesenchymal Stem Cells Isolated from Immunodeficient Mice

**DOI:** 10.3390/ijms20112672

**Published:** 2019-05-30

**Authors:** Yoshiki Nakashima, Saifun Nahar, Chika Miyagi-Shiohira, Takao Kinjo, Naoya Kobayashi, Shinji Kitamura, Issei Saitoh, Masami Watanabe, Jiro Fujita, Hirofumi Noguchi

**Affiliations:** 1Department of Regenerative Medicine, Graduate School of Medicine, University of the Ryukyus, Okinawa 903-0215, Japan; nakasima@med.u-ryukyu.ac.jp (Y.N.); chika@med.u-ryukyu.ac.jp (C.M.-S.); 2Department of Infectious, Respiratory, and Digestive Medicine, Graduate School of Medicine, University of the Ryukyus, Okinawa 903-0215, Japan; tuliferdous@yahoo.com (S.N.); fujita@med.u-ryukyu.ac.jp (J.F.); 3Department of Tropical Medicine and Parasitology, Faculty of Medicine, Juntendo University, Tokyo 113-8421, Japan; 4Department of Basic Laboratory Sciences, School of Health Sciences in the Faculty of Medicine, University of the Ryukyus, Okinawa 903-0215, Japan; kinjotko@med.u-ryukyu.ac.jp; 5Okayama Saidaiji Hospital, Okayama 704-8192, Japan; n-kobayashi@saidaiji-hp.or.jp; 6Department of Nephrology, Rheumatology, Endocrinology and Metabolism, Okayama University Graduate School of Medicine, Dentistry and Pharmaceutical Sciences, Okayama 700-8558, Japan; kitamura@okayama-u.ac.jp; 7Division of Pediatric Dentistry, Graduate School of Medical and Dental Science, Niigata University, Niigata 951-8514, Japan; isaito@dent.niigata-u.ac.jp; 8Department of Urology, Okayama University Graduate School of Medicine, Dentistry and Pharmaceutical Sciences, Okayama 700-8558, Japan; masami5@md.okayama-u.ac.jp

**Keywords:** adult stem cells, mesenchymal stem cell, tissue regeneration, regenerative medicine

## Abstract

Although cell therapy using adipose-derived mesenchymal stem cells (AdMSCs) regulates immunity, the degree to which cell quality and function are affected by differences in immunodeficiency of donors is unknown. We used liquid chromatography tandem-mass spectrometry (LC MS/MS) to identify the proteins expressed by mouse AdMSCs (mAsMSCs) isolated from normal (C57BL/6) mice and mice with severe combined immunodeficiency (SCID). The protein expression profiles of each strain were 98%–100% identical, indicating that the expression levels of major proteins potentially associated with the therapeutic effects of mAdMSCs were highly similar. Further, comparable levels of cell surface markers (CD44, CD90.2) were detected using flow cytometry or LC MS/MS. MYH9, ACTN1, CANX, GPI, TPM1, EPRS, ITGB1, ANXA3, CNN2, MAPK1, PSME2, CTPS1, OTUB1, PSMB6, HMGB1, RPS19, SEC61A1, CTNNB1, GLO1, RPL22, PSMA2, SYNCRIP, PRDX3, SAMHD1, TCAF2, MAPK3, RPS24, and MYO1E, which are associated with immunity, were expressed at higher levels by the SCID mAdMSCs compared with the C57BL/6 mAdMSCs. In contrast, ANXA9, PCBP2, LGALS3, PPP1R14B, and PSMA6, which are also associated with immunity, were more highly expressed by C57BL/6 mAdMSCs than SCID mAdMSCs. These findings implicate these two sets of proteins in the pathogenesis and maintenance of immunodeficiency.

## 1. Introduction 

Stem cells play an important role in maintaining tissue homeostasis. Mesenchymal stem cells (MSCs) [1] have drawn particular attention for application to regenerative medicine [2]. MSCs reside in human bone marrow [3], umbilical cord [4], synovial membrane [5], dental pulp [6], and adipose tissue [7], and differentiate into the somatic cells of bone, cartilage, and fat tissues. Tissue regeneration therapy using MSCs has been investigated in research and clinical settings for many years. Moreover, the immunomodulatory functions of MSCs have also garnered attention. For example, MSC transplantation is used to treat autoimmune diseases [8,9] and inflammatory arthritis [10,11]. While numerous clinical reports describe the utility of bone marrow-derived MSCs, the collection of bone marrow is difficult for patients. In contrast, adipose-derived MSCs (AdMSCs) are particularly useful, because they can be easily collected from a patient’s subcutaneous fat [12,13,14] for use in autologous transplantation. Moreover, autologous cell therapy can be easily performed without immunological rejection. 

Evidence indicates that AdMSCs are involved in the regulation of immunity [15,16] through, for example, the secretion of growth factors [16]. Cellular therapy using AdMSCs is expected to reduce the symptoms of autoimmune diseases such as rheumatoid arthritis [17] and dermatitis [9]. Further, AdMSCs have potential therapeutic effects that may alleviate the symptoms of immune system dysfunction caused by exposure to cosmic radiation, exposure to accidental release of ionizing radiation by nuclear power plants as well as by viral infections. However, information is lacking about whether cell quality and the effects of cell therapy are equivalent between AdMSCs collected from healthy donors or from donors with diseases affecting the immune system. We must therefore determine whether autologous or allogeneic transplantation is optimal when cells are provided by patients with diseases of the immune system. CD4+ T cells play a central role in cellular immunity [18], whereas B cells produce antibodies [19] and play a central role in humoral immunity [20]. It was also reported that HIV inactivated B cells in addition to T cells [21]. In general, cells of the immune system are susceptible to radiation exposure, and T and B cells are particularly sensitive [22].

To determine whether patients suffering from immunodeficiency diseases should use autologous AdMSCs, we collected AdMSCs from mice with severe combined immunodeficiency (SCID) and characterized their phenotypes. SCID mice are spontaneous mutants that were first identified in C. B-17 strain mice in 1980 by Bosma et al. [23]. The color of these mice is determined by a stereotypical recessive pattern of inheritance. These mice lack B and T cells, which accounts for their severe immunodeficiency, and makes it unlikely that they will reject transplanted heterologous cells or tissues. Moreover, SCID mice serve as a useful model for studying immunodeficiency caused by HIV infection [24] or exposure to ionizing radiation [25].

Here, to better understand whether the phenotypes of AdMSCs isolated from patients suffering from immune diseases are suitable for therapy, we compared the proliferation, differentiation, surface marker expression, and protein expression of mAdMSCs of SCID and normal (C57BL/6) mice. We then analyzed proteins expressed by mAdMSCs using liquid chromatography tandem-mass spectrometry (LC-MS/MS). Finally, we classified the functions of proteins identified using Gene Ontology analysis (GO) [26,27], with a focus on proteins differentially expressed by these mouse strains that are associated with immune response, viral response, B cells, and T cells.

## 2. Results

### 2.1. Characteristics and Quality of C57BL/6 and SCID mAdMSCs

The mAdMSCs were cultured to 80% confluence in DMEM medium containing 10% FBS, with the medium exchanged every two days. Microscopy was performed to confirm that there were no abnormalities in the sizes or shapes of the C57BL/6 mAdMSCs (Figure 1A, left panel) and SCID mAdMSCs (Figure 1A, right panel) or in culture conditions. Flow cytometry was performed to detect markers of mAdMSCs (CD44 and CD90.2), hematopoietic stem cells (CD34), and leukocytes (CD45). mAdMSCs expressed CD44 and CD90.2 but not CD34 and CD45 (Figure 1B, left panels: C57BL/6, right panels: SCID). We induced adipocyte differentiation of the C57BL/6 mAdMSCs (Figure 1C, upper-left panel) and SCID mAdMSCs (Figure 1C, upper-middle panel) and osteoblasts (Figure 1C, upper-right panel). We induced osteocytic differentiation of the C57BL/6 mAdMSCs (Figure 1C, lower-left panel) and SCID mAdMSCs (Figure 1C, lower-middle panel) mAdMSCs and the differentiation of osteoblasts (Figure 1C, lower-right panel) into osteocytes. Mature adipocytes and osteocytes were stained with Oil Red O and Alizarin Red S, respectively.

### 2.2. Analysis of Protein Expression by C57BL/6 and SCID mAdMSCs

Appendix A list the 1097 proteins expressed by C57BL/6 mAdMSCs (n = 1097) and SCID mAdMSCs (n = 978). Values of the exponentially modified protein abundance indexes (emPAIs) are shown in the right columns of each. We previously reported that emPAI >10 is a reliable detection setting when a single sample is analyzed [28]. In the present study, we evaluated the numbers of proteins detected at emPAI values >0 (Figure 2A), >5 (Figure 2B), and >10 (Figure 2C).

### 2.3. Analysis of Proteins Expressed by C57BL/6 mAdMSCs and SCID mAdMSCs (emPAI > 0)

C57BL/6 mAdMSCs and SCID mADMSCs expressed 1688 and 1463 proteins, respectively, among which 398 (22%) were unique to C57BL/6 mAdMSCs. Both types of mAdMSCs expressed the same set of 1290 proteins, and 173 (9%) of the 1463 proteins were uniquely expressed by SCID mAdMSCs (Figure 2A).

### 2.4. Analysis of Proteins Expressed by C57BL/6 mAdMSCs and SCID mAdMSCs (emPAI > 5)

C57BL/6 mAdMSCs and SCID mAdMSCs expressed 1022 and 1029 proteins, respectively. Of the 1022 proteins expressed by C57BL/6 mAdMSCs, 9 (1%) were unique. C57BL/6 mAdMSCs and SCID mAdMSCs expressed the same set of 1013 proteins. Among the 1029 proteins expressed by SCID mAdMSCs, 16 (1%) were unique (Figure 2B).

### 2.5. Analysis of Proteins Expressed by C57BL/6 mAdMSCs and SCID mAdMSCs (emPAI > 10)

C57BL/6 mAdMSCs and SCID mAdMSCs expressed 630 proteins, among which 1 (0%) was unique to C57BL/6 mAdMSCs. Both types of mAdMSCs expressed the same set of 628 proteins, and 2 (0%) were specific to SCID mAdMSCs (Figure 2C).

### 2.6. Analysis of Proteins Expressed by C57BL/6 mAdMSCs and SCID mAdMSCs (emPAI > 5)

C57BL/6 mAdMSCs and SCID mAdMSCs expressed 1022 and 1029 proteins, respectively. Nine of the 1022 (1%) proteins were uniquely expressed by C57BL/6 mAdMSCs (myristoylated alanine-rich C-kinase substrate [MARCKS], sterol-4- alpha-carboxylate 3-dehydrogenase, decarboxylating [NSDHL], ferritin light chain 1 [FTL1], tripeptidyl- peptidase 2 [TPP2], cytosolic acyl coenzyme A thioester hydrolase [ACOT7], cellular retinoic acid-binding protein 1 [CRABP1], phospholipid phosphatase 3 [PLPP3], thrombomodulin [THBD], and peptidyl-prolyl cis-trans isomerase FKBP4 [FKBP4]). Both types of mAdMSCs expressed the same set of 1029 proteins, and 16 were uniquely expressed by SCID mAdMSCs (cAMP-dependent protein kinase type II-beta regulatory subunit [PRKAR2B], pre-B-cell leukemia transcription factor-interacting protein 1 [PBXIP1], growth factor receptor-bound protein 10 [GRB10], nucleobindin-2 [NUCB2], nuclear factor 1 C-type [NFIC], Ras- related GTP-binding protein C [RRAGC], CLIP-associating protein 2 [CLASP2], SH3 domain-containing kinase-binding protein 1 [SH3KBP1], translin-associated protein X [TSNAX], prolyl 4-hydroxylase subunit alpha-3 [P4HA3], CDGSH iron-sulfur domain-containing protein 1 [CISD1], glucosamine 6-phosphate N-acetyltransferase [GNPNAT1], amine oxidase A [MAOA], fatty aldehyde dehydrogenase [ALDH3A2], synaptophysin-like protein 1 [SYPL1], and RNA-binding protein with multiple splicing [RBPMS]). Thus, 98% of the proteins detected were expressed by both types of mAdMSCs (Figure 2B).

### 2.7. Analysis of Proteins Expressed by C57BL/6 mAdMSCs and SCID mAdMSCs (emPAI > 10)

C57BL/6 mAdMSCs and SCID mAdMSCs expressed 629 and 630 proteins, respectively, among which 1 (0%) (MARCKS) was uniquely expressed by C57BL/6 mAdMSCs. The SCID mAdMSCs uniquely expressed PRKAR2B and PBXIP1 (Figure 2C).

### 2.8. GO Classification (emPAI > 0)

The proteins were classified according to the subcategories of the Gene Ontology Consortium (available online: http://www.geneontology.org/) database as follows: biological processes, cellular components, and molecular functions. The proteins uniquely expressed by C57BL/6 mAdMSCs and SCID mAdMSCs are shown in Figure 3, left panel. The condition emPAI >0 identified 60 and 53 “viral process-related proteins” expressed by C57BL/6 mAdMSCs and SCID mAdMSCs, respectively. The condition emPAI >0 identified 154 and 137 “immune system process-related proteins” expressed by C57BL/6 mAdMSCs and SCID mAdMSCs, respectively.

### 2.9. GO Classification (emPAI > 5)

The proteins uniquely expressed by C57BL/6 mAdMSCs and SCID mAdMSCs, respectively, are shown in Figure 3. The condition emPAI >5 identified 44 “viral process-related proteins,” each expressed by C57BL/6 mAdMSCs and SCID mAdMSCs. The condition emPAI >0 identified 110 and 109 “immune system process-related proteins” expressed by C57BL/6 mAdMSCs and SCID mAdMSCs, respectively.

### 2.10. GO Classification (emPAI > 10 and emPAI > 0)

The proteins uniquely expressed by C57BL/6 mAdMSCs and SCID mAdMSCs are shown in (Figure 3). The condition emPAI >10 identified 28 “viral process-related proteins” each expressed by C57BL/6 mAdMSCs SCID mAdMSCs. The condition emPAI >0 identified 75 and 76 “immune system process-related proteins” expressed by C57BL/6 mAdMSCs and SCID mAdMSCs, respectively.

### 2.11. B Cell-Related Proteins Expressed by C57BL/6 mAdMSCs and SCID mAdMSCs

The 13 B cell-related proteins detected were as follows: 60 kDa heat shock protein [HSPD1], Mitogen-activated protein kinase 1 [MAPK1], Purine nucleoside phosphorylase [PNP], CTP synthase 1 [CTPS1], Macrophage migration inhibitory factor [MIF], Transferrin receptor protein 1 [TFRC], Matrix metalloproteinase-14 [MMP14], CD81 antigen [CD81], CD44 antigen [CD44], Caspase-3 [CASP3], Signal transducer and activator of transcription 5A [STAT5A], Apoptosis regulator BAX [BAX], and Aurora kinase B [AURKB]). Among them, eight were expressed by SCID mAdMSCs. The emPAI values were normalized according to the levels of proteins encoded by housekeeping genes (HKGs). MAPK1 and CTPS1 were highly expressed by SCID mAdMSCs. HSPD1, PNP, MIF, TFRC, and MMP14 were expressed at comparable levels by C57BL/6 mAdMSCs and SCID mAdMSCs. Low levels of CD81, CD44, STAT5A, BAX, and AURKB were uniquely expressed by C57BL/6 mAdMSCs (Appendix A).

### 2.12. T Cell-Related Proteins Expressed by C57BL/6 mAdMSCs and SCID mAdMSCs

The 43 T cell-related proteins detected were as follows: Cluster of Heat shock protein HSP 90-beta [HSP90AB1], Heat shock protein HSP 90-alpha [Hsp90aa1], Cluster of Myosin-9 [MYH9], 60 kDa heat shock protein [HSPD1], Gelsolin [GSN], Annexin A1 [ANXA1], Transforming protein RhoA [RHOA], Mitogen-activated protein kinase 1 [MAPK1], RAC-alpha serine/threonine-protein kinase [AKT1], Heat shock protein 105 kDa [HSPH1], Galectin-3 [LGALS3], Purine nucleoside phosphorylase [PNP], CTP synthase 1 [CTPS1], High mobility group protein B1 [HMGB1], Integrin alpha-V [ITGAV], Amyloid beta A4 precursor protein-binding family B member 1-interacting protein [APBB1IP], Catenin beta-1 [CTNNB1], 60S ribosomal protein L22 [RPL22], Transferrin receptor protein [TFRC], Peptidyl-prolyl cis-trans isomerase FKBP1A [FKBP1A ], Peroxiredoxin-2 [PRDX2], Thy-1 membrane glycoprotein [THY1], Vascular cell adhesion protein 1 [EZR], Disks large homolog 1 [DLG1], Ubiquitin-conjugating enzyme E2 N [UBE2N], Proteasomal ubiquitin receptor ADRM1 [ADRM1], cAMP-dependent protein kinase type I-alpha regulatory subunit [PRKAR1A], 40S ribosomal protein S6 [RPS6], CD44, AP-3 complex subunit delta-1 [AP3D1], Dual specificity protein phosphatase 3 [DUSP3], Beta-2-microglobulin [B2M], Caveolin-1 [CAV1], Deoxyhypusine synthase [DHPS], Caspase-3 [CASP3], Cluster of Tyrosine-protein kinase CSK [CSK], Signal transducer and activator of transcription 5A [STAT5A], WASH complex subunit 1 [WASHC1], Pro-cathepsin H [CTSH], Cleft lip and palate transmembrane protein 1 homolog [CLPTM1], Apoptosis regulator BAX [BAX], and Serine/threonine-protein kinase D2 [PRKD2]). Among them, 37 were uniquely expressed by SCID mAdMSCs. The emPAI values were normalized according to those of proteins encoded by HKGs. MYH9, MAPK1, and CTPS1 were highly expressed in SCID mAdMSCs. HSP90AB1, HSP90AA1, HSPD1, GSN, ANXA1, RHOA, AKT1, HSPH1, LGALS3, PNP, HMGB1, ITGAV, APBB1IP, CTNNB1, RPL22, TFRC, FKBP1A, PRDX2, THY1, EZR, VCAM1, DLG1, UBE2N, PRKAR1A, RPS6, DUSP3, B2M, CAV1, CASP3 and CLPTM1 were detected at comparable levels in C57BL/6 mAdMSCs and SCID mAdMSCs. Low levels of ADRM1, CD44, AP3D1, STAT5A, CTSH and BAX were only detected in the C57BL/6 mAdMSCs, and low levels of DHPS, CSK, WASHC1, and PRKD2 were detected in the SCID mAdMSCs (Appendix A).

### 2.13. Viral Process-Related Proteins Expressed by C57BL/6 mAdMSCs and SCID mAdMSCs

Viral process-related proteins were as follows: Heat shock protein HSP 90-beta [HSP90AB1], Heat shock cognate 71 kDa protein [Hspa8], Alpha-enolase [ENO1], TALIN-1 [TLN1], Transitional endoplasmic reticulum ATPase [VCP], Elongation factor 1-gamma [EEF1G], T-complex protein 1 subunit epsilon [CCT5], Cathepsin B [CTSB], ATP-dependent RNA helicase DDX3X [DDX3X], E3 ubiquitin-protein ligase NEDD4 [NEDD4], Spliceosome RNA helicase DDX39B [DDX39B], Eukaryotic translation initiation factor 3 subunit B [EIF3B], 60S ribosomal protein L3 [RPL3], Eukaryotic translation initiation factor 3 subunit A [EIF3A], 40S ribosomal protein S15a [RPS15A], ATP-dependent RNA helicase DDX1 [DDX1], Programmed cell death 6-interacting protein [PDCD6IP], Poly(rC)-binding protein 2 [PCBP2], Heat shock protein beta-1 [HSPB1], Polypyrimidine tract-binding protein 1 [PTBP1], Nuclear pore complex protein Nup155 [NUP155], E3 ubiquitin-protein ligase UBR4 [UBR4], Eukaryotic peptide chain release factor subunit 1 [ETF1], Ras-related protein Rab-7a [RAB7A], Integrin alpha-V [ITGAV], Poly(rC)-binding protein 1 [PCBP1], Deoxynucleoside triphosphate triphosphohydrolase SAMHD1 [SAMHD1] and Nucleolar RNA helicase 2 [DDX21]). The 28 proteins were expressed by C57BL/6 mAdMSCs and SCID mAdMSCs. The emPAI values were corrected according to the expression of the HKG-encoded proteins. EEF1G, DDX39B, RPS15A, DDX1, NUP155, and SAMHD1 associated with the viral process were highly expressed in SCID mAdMSCs. The other proteins were detected at comparable levels (Appendix A).

### 2.14. Immune System Process-Related Proteins Expressed by C57BL/6 mAdMSCs and SCID mAdMSCs

The 70 immune system process-related proteins were as follows: (Cluster of Heat shock protein HSP 90-beta [HSP90AB1], Myosin-9 [MYH9], Alpha-actinin-1 [ACTN1], Glyceraldehyde-3-phosphate dehydrogenase [GAPDH], Elongation factor 2 [EEF2], Annexin A2 [ANXA2], Stress-70 protein [HSPA9], 60 kDa heat shock protein [HSPD1], WD repeat-containing protein 1 [WDR1], Protein disulfide-isomerase A3 [PDIA3], Moesin [MSN], Calreticulin [CALR], Spectrin beta chain, non-erythrocytic 1 [SPTBN1], Galectin-1 [LGALS1], Annexin A1 [ANXA1], ATP-dependent RNA helicase DDX3X [DDX3X], Calnexin [CANX], Glucose-6-phosphate isomerase [GPI], Tropomyosin alpha-1 chain [TPM1], Bifunctional glutamate/proline--tRNA ligase [EPRS], Importin-7 [IPO7], Transforming protein RhoA [RHOA], 40S ribosomal protein S7 [RPS7], Integrin beta-1 [ITGB1], E3 ubiquitin-protein ligase NEDD4 [NEDD4], Annexin A3 [ANXA3], Peroxiredoxin-1 [PRDX1], Proteasome subunit beta type-2 [PSMB2], Calponin-2 [CNN2], Dolichyl-diphosphooligosaccharide--protein glycosyltransferase 48 kDa subunit [DDOST], Mitogen-activated protein kinase 1 [MAPK1], Mitogen-activated protein kinase 1 [PSMA4], Proteasome subunit beta type-1 [PSMB1], Proteasome activator complex subunit 2 [PSME2], RAC-alpha serine/threonine-protein kinase [AKT1], CTP synthase 1 [CTPS1], Poly(rC)-binding protein 2 [PCBP2], Complement component 1 Q subcomponent-binding protein [C1QBP], Ubiquitin thioesterase OTUB1 [OTUB1], Switch-associated protein 70 [SWAP70], Galectin-3 [LGALS3], Tyrosine- protein phosphatase non-receptor type 11 [PTPN11], Proteasome subunit alpha type-5 [PSMA5], Superoxide dismutase [SOD2], Proteasome subunit beta type-6 [PSMB6], Macrophage migration inhibitory factor [MIF], High mobility group protein B1 [HMGB1], 40S ribosomal protein S19 [RPS19], Integrin alpha-V [ITGAV], Proteasome subunit alpha type-7 [PSMA7], Proteasome subunit beta type-3 [PSMB3], Protein transport protein Sec61 subunit alpha isoform 1 [SEC61A1], Amyloid beta A4 precursor protein-binding family B member 1-interacting protein [APBB1IP], Catenin beta-1 [CTNNB1], Lactoylglutathione lyase [GLO1], 60S ribosomal protein L22 [RPL22], Protein phosphatase 1 regulatory subunit 14B [PPP1R14B], Dual specificity mitogen-activated protein kinase kinase 1 [MAP2K1], Proteasome subunit alpha type-1 [PSMA1], Proteasome subunit alpha type-2 [PSMA2], Proteasome subunit alpha type-6 [PSMA6], Transferrin receptor protein 1 [TFRC], Cytoplasmic dynein 1 light intermediate chain 1 OS [DYNC1LI1], Heterogeneous nuclear ribonucleoprotein Q [SYNCRIP], Thioredoxin-dependent peroxide reductase [PRDX3], Deoxynucleoside triphosphate triphosphohydrolase SAMHD1 [SAMHD1], TRPM8 channel-associated factor 2 [TCAF2], Mitogen- activated protein kinase 3 [MAPK3], 40S ribosomal protein S24 [RPS24], and Unconventional myosin-Ie [MYO1E]). The emPAI values were corrected according to expression of the HKG-encoded proteins. MYH9, ACTN1, CANX, GPI, TPM1, EPRS, ITGB1, ANXA3, CNN2, MAPK1, PSME2, CTPS1, OTUB1, PSMB6, HMGB1, RPS19, SEC61A1, CTNNB1, GLO1, RPL22, PSMA2, SYNCRIP, PRDX3, SAMHD1, TCAF2, MAPK3, RPS24, and MYO1E were highly expressed in SCID mAdMSCs. ANXA9, PCBP2, LGALS3, PPP1R14B, and PSMA6 were highly expressed in C57BL/6 mAdMSCs. The other proteins were detected at comparable levels in C57BL/6 mAdMSCs and SCID mAdMSCs (Appendix A).

### 2.15. Effects of Culture Conditions on Protein Expression by AdMSCs

Immunofluorescence analysis revealed that SCID mAdMSCs expressed CD44 and CD90.2 at normal levels (Figure 4A), consistent with the results of flow cytometry (Figure 1B). The proliferation of C57BL/6 and SCID mAdMSCs isolated from female mice was assessed. The amounts of DNA obtained from C57BL/6 mAdMSCs and SCID mAdMSCs 1, 3, and 5 days after cell seeding were equivalent (Figure 4B). Further, SCID mAdMSCs isolated from female and male mice expressed mRNAs encoding CD44 and CD90.2 at normal levels (Figure 4C).

The influence of the presence or absence of a scaffold material (gelatin-coat) on the function of mAdMSCs was investigated. The C57BL/6 mAdMSCs and SCID mAdMSCs expressed equivalent levels of mRNAs encoding the growth factors HGF, VEGF, and TGFB (Figure 4D, upper panel) and the antioxidant factors HO-1 and iNOS (Figure 4D, lower panel) when cultured with or without a gelatin coating.

The influence of the presence or absence of an inflammatory environment on the function of mAdMSCs was also investigated. The levels of growth factors HGF, VEGF, TGFB and antioxidant factors HO-1, iNOS expressed by C57BL/6 mAdMSCs and SCID mAdMSCs were not affected by the addition of LPS to the culture medium.

### 2.16. Quantitative Analysis of Proteins Expressed by C57BL/6 mAD MSCs and SCID mAdMSCs

The normalized levels of proteins expressed by C57BL/6 mAD MSCs and SCID mAdMSCs are shown in Figure 5. The average normalized level of proteins expressed by C57BL/6 mAdMSCs was 106.0% of that of SCID mAdMSC (HKGs: YWHAZ, GAPDH, ATP5F1B, PGK1, PPIA, and TFRC) (Appendix A). The values of FLNA, SERPINH1, PKM, ANXA2, S100A11, PGAM1, VDAC1, A100A6, TKT, and HSPA9 of C57BL/6 mAdMSCs were higher compared with those of SCID mAdMSCs. The values of MYH9, ACTG2, TUBB5, ACTN1, CLTC, DYNC1H1, HSP90B1, TUBA4A, PLS3, MYH10, MYH11, and MYO1C of SCID mAdMSCs were higher compared with those of C57BL/6 mAdMSCs (Figure 5).

### 2.17. Correlation Analysis of Proteins Expressed by mAdMSCs

The normalized values of the proteins expressed by C57BL/6 mAdMSCs and SCID mAdMSCs are shown in Appendix A. The average value of proteins expressed by C57BL/6 mAdMSCs was 106.0% of those expressed by C57BL/6 SCID mAdMSCs. (HKGs: YWHAZ, GAPDH, ATP5F1B, PGK1, PPIA, and TFRC) (Appendix A). The values of ANXA2 and HSPA9 expressed by C57BL/6 mAdMSCs were higher compared with those of SCID mAdMSCs. The values of MYH9 and ACTN1 expressed by SCID mAdMSCs were higher compared with those of C57BL/6 mAdMSCs (Appendix A).

## 3. Discussion

### 3.1. Reliability of LC-MS/MS for Comparative Analysis of Protein Expression

Here, we employed LC-MS/MS to conduct a comparative analysis of proteins expressed by mAdMSCs isolated from SCID and normal mice. The ratio of the number of measured peptides to the number of theoretical peptides was linearly related to the logarithm of the protein concentration using the emPAI, which is defined by subtracting 1 from the index of the peptide-number ratio. The larger the emPAI value, the greater the number of proteins. The LC-MS/MS shotgun analysis detected peptides whose origins were determined using an online protein database. Here we combined inguinal-pad fat harvested from three C57BL/6 mice and three SCID mice to produce single cultures of mAdMSCs from each source. The reason for this approach is that unlike the case in humans, the amount of adipose tissue that can be extracted in mice is very small. We recently published a correlation between emPAI values (>0, >1, >2, >3, >5, >10) and the results of protein expression analyses. For example, using emPAI >10, a protein can be detected with high probability even when using n = 1 samples [20]. Thus, we concluded that protein expression data generated using a high emPAI (emPAI >10) is reliable. However, the data of these “n = 1” experiments was strongly affected by individual differences. Thus, the LC-MS/MS protein expression data presented here reveal that the individual differences among SCID mice were consistent with those of normal mice. This is a pilot study of protein expression, and it should be noted therefore that the reliability of the data is influenced by the research method.

### 3.2. T Cell/B Cell-Associated Proteins

Here we show that cultured AdMSCs expressed the B cell- and T cell-associated proteins (MAPK1 and CTPS1). MAPK1, also known as extracellular signal-regulated kinase 2 (ERK2), phosphorylates proteins in the cytoplasm and nucleus and is involved in diverse cell functions, ranging from cell cycle control to higher brain functions. ERK2 promotes T cell activation [29] and the proliferation and survival of CD8-positive T cells [30]. However, the increase in the levels of MAPK1, which acts in the intracellular signaling pathway of AdMSCs, does not mean that AdMSCs directly increase the expression of MAPK1 in B and T cells. For example, human bone marrow-derived MSCs are likely to differentiate into adipocytes when ERK1 and ERK2 are activated. However, in our experiments, induction of the differentiation of AdMSCs into adipocytes did not differ between AdMSCs derived from SCID mice or C57BL/6 mice (Figure 3C).

In contrast, the loss of function of Cts1 is selectively impaired by T cell proliferation after antigen stimulation. Thus, CTS1 is an important checkpoint in adaptive immunity [31]. Further, a loss-of-function homozygous mutation of Ctps1 inhibits the proliferation of T and B cells in response to activation by an antigen receptor [32]. We were unable to identify studies of the expression or function of CTS1 by MSCs. MSCs and B cells, when cocultured, regulate IgG secretion [33], the inhibition of apoptosis [34], proliferation [35], inhibition of plasma cell differentiation [36], and chemotaxis of B cells [37]. However, the mechanisms of many of these activities, such as those that involve cell adhesion as well as the role of humoral factors, are unknown [38].

### 3.3. Global Comparison of Proteins Expressed by mAdMSCs

After normalizing the levels of proteins expressed by mAdMSCs isolated from the two mouse strains to those encoded by HKGs, we found that the average level of proteins expressed by C57BL/6 mAdMSCs was 1.06-times higher compared with that of the SCID mAdMSCs. Thus, the mean normalized value of total protein levels of the SCID mAdMSCs was 122.0% that of the C57BL/6 mAdMSCs (Figure 5).

The growth rates of the mADMSCs isolated from the two mouse strains were similar (Figure 4B) and did not change when the culture dishes were coated with gelatin or when the cultures were treated with LPS. Thus, it is unclear why the average amount of protein expressed in SCID mAdMSCs was approximately 20% higher compared with that of normal mouse mAdMSCs. This may be explained by differences in the genotypes of the two mouse strains.

### 3.4. Proteins Related to the Immune Response

The 165 proteins associated with the GO term “immune system process” of biological processes were investigated (Figure 3). After normalization using HKGs, the emPAI value of the myosin subunit MYH9 was higher for SCID mAdMSCs compared with that of the C57BL/6 mAdMSCs (Appendix A). Moreover, the mean level of immunity-associated proteins after normalization was 121.0% that of the C57BL/6 group (Appendix A). However, proteins involved in specific immune responses were not detected.

### 3.5. Analysis of Proteins Associated with the Viral Response

Next, we focused on the expression of proteins related to the GO terms viral response (Appendix A) and immune-related proteins (Appendix A). The levels of viral process-related protein factors (SAMHD1) and immune-related protein factors (CTPS1, HMGB1 and SAMHD1) were high in SCID mAdMSCs. SAMHD1 protects cells from viral infection and is involved in the development of cancer and chronic inflammation. Further, the mechanism was recently identified [39]. Further, loss-of-function mutations in *Ctps1* are associated with the severe and selective impairment of T cell proliferation after antigen stimulation, indicating that CTPS1 serves as a checkpoint in adaptive immunity [31]. The cytokine HMGB1, which is produced in response to injury, infection, and inflammatory stimuli, is secreted by activated macrophages, mature dendritic cells, and natural killer cells [40]. 

### 3.6. Summary

The results of studies using mice with severe immune disorders that are deficient in T and B cells, such as SCID mice, can be used as references for evaluating the function of MSCs derived from patients with immunological abnormalities. In the present study, the constellations of proteins and their levels were highly similar between MSCs isolated from normal SCID mice. Moreover, we compared the levels of proteins that mediate diverse biological functions between mAdMSCs generated from three SCID mice compared with those expressed by mAdMSCs generated from three C57BL/6 mice. The levels of representative growth factors and cell surface markers were comparable between SCID and C57BL/6 mAdMSCs. Further, 28 GO-category viral response-related factors and 70 GO-category immune-related factors were expressed by SCID mAdMSCs and C57BL/6 mAdMSCs. However, MYH9, ACTN1, CANX, GPI, TPM1, EPRS, ITGB1, ANXA3, CNN2, MAPK1, PSME2, CTPS1, OTUB1, PSMB6, HMGB1, RPS19, SEC61A1, CTNNB1, GLO1, RPL22, PSMA2, SYNCRIP, PRDX3, SAMHD1, TCAF2, MAPK3, RPS24, and MYO1E, which are associated with immunity, were more highly expressed by SCID mAdMSCs compared with those of C57BL/6 mAdMSCs. In contrast, ANXA9, PCBP2, LGALS3, PPP1R14B, and PSMA6, which are associated with immunity, were more highly expressed in C57BL/6 mAdMSCs than in SCID mAdMSCs. In particular, SCID mAdMSCs expressed the immunity-associated protein SAMHD1 as well as CTPS1, HMGB1, and SAMHD1 at higher levels compared with those of C57BL/6 mAdMSCs. However, no studies are available that identify the roles these proteins play in the therapeutic effects of AdMSCs. The results of the present study provide important clues that may help guide the selection of targets when investigating the underlying mechanisms of the regulation of the immune response by AdMSCs.

## 4. Materials and Methods

### 4.1. Reagents and Materials 

Fetal bovine serum (FBS) was obtained from BioWest (Nuaille, France). D-MEM (high glucose) with L-glutamine, phenol red, and sodium pyruvate (DMEM) were obtained from Fujifilm Wako Pure Chemical Corporation (Osaka, Japan). Mouse osteoblasts from cranial bone (Code No. OBC12C) were obtained from Cosmo Bio Co., Ltd. (Tokyo, Japan). Plastic dishes were obtained from TPP (Trasadingen, Switzerland). All other materials used were of the highest commercial grade.

### 4.2. Animal Care

Experimental protocols were conducted in accordance with the guidelines for the care and use of laboratory animals published by the Research Laboratory Center, Faculty of Medicine and the Institute for Animal Experiments, Faculty of Medicine, University of the Ryukyus (Okinawa, Japan). The experimental protocol was approved by the Committee on Animal Experiments of University of the Ryukyus (permit number: R2017081. 08/05/2018). C57BL/6 female mice (8 weeks of age; Japan SLC, Shizuoka, Japan) and CB17/Icr-Prkdc^scid^/CrlCrlj male and female mice (8 weeks of age; Charles River Laboratories Japan, Inc., Kanagawa, Japan) were maintained under controlled temperature (23 ± 2 °C) and illumination conditions (lights on from 08:30–20:30). Animals were fed standard rodent chow pellets and had free access to water. All efforts were made to minimize the suffering of the animals.

### 4.3. Isolation of AdMSCs from Mouse Adipose Tissue Via the Inguinal Pad Fat

Adipose tissue was obtained from the inguinal pad fat of three 8-week-old mice (female, n = 3). The method for isolating AdMSCSs from adipose tissue was in accordance with the adipose tissue-derived stem cell product standard document (RMRC-A 01: 2015) of the Ryukyus Regenerative Medicine Research Center. Tissues were stored in cold HBSS and vigorously washed three times in HBSS before starting digestion. Next, the tissues were cut into small fragments with a scalpel for enzymatic digestion (2 mg collagenase type IV/ml; HBSS) in 50-ml tubes rotated at 20 RPM, 37 °C for 60 min) using a shaker (BioShaker BR-42FM; TAITEC, Saitama, Japan). The tubes were then centrifuged at 800× *g* for 5 min. The SVF [41] containing multiple cell types, including AdMSCs, was collected from the bottom of the tube after centrifugation. The AdMSCs were collected as a cell pellet and then washed with fresh DMEM medium containing 10% FBS to remove the enzyme after digestion. The digested tissue was then incubated in a T25 flask.

### 4.4. Preparation of mAdMSCSs

Mouse AdMSCSs were cultured (37 °C, 5% CO_2_) in an uncoated T25 flask (TPP 90026). Cells were passaged every 3 to 4 days after seeding until reaching 80% confluence. The cells were then washed with PBS (calcium, magnesium-free), and mouse AdMSCs were dissociated using Trypsin/EDTA (Lonza CC-3232). Cells were subcultured in an uncoated T25 flask containing DMEM medium supplemented with 10% FBS. The mAdMSCs were passaged three times.

### 4.5. Real-Time PCR (qPCR)

Cells (1 × 10^4^) were seeded onto 24-well plates. Subconfluent cells were dissociated with trypsin - EDTA and recovered. RNA was prepared for a qPCR using an RNeasy Mini Kit and QuantiTect Reverse Transcription Kit according to the manufacturer’s instructions (QIAGEN, Hilden, Germany). Reactions were performed using a LightCycler 96 Real-Time PCR system (Roche, Basel, Switzerland). The FastStart Essential DAN Green Master (Roche) was used according to the manufacturer’s instructions.

The primers used for the real-Time PCR were as follows:
mouse CD44 forward, TACTCATATTCTAGCCTCCCTCCTT,mouse CD44 reverse, GTGGAGAATAGCCAAGAATCATCTA,mouse CD90.2 forward, GTCCTTCAAATATCTCAGAACATGG,mouse CD90.2 reverse, GCCCTGGAATAAATACAGAGTACAA,mouse HGF forward, ACTCTTACCAAGGAAGACCCATTAC,mouse HGF reverse, ATACCAGTAGCATCGTTTTCTTGAC,mouse VEGF forward, TGTCTTCACTGGATATGTTTGACTG,mouse VEGF reverse, TTCTCTGTCATCATCTGTCTCTCTG,mouse TGF-β forward, AGTAGCTCCCCTATTTAAGAACACC,mouse TGF-β reverse, GGAAAGGTAGGTGATAGTCCTGAAT,mouse HO-1 forward, TCCAGACATTTCTGTCTCGTATTTC,mouse HO-1 reverse, CACACAGAAGTTAGAGACCAAGGTT,mouse iNOS forward, AGCTTCTGGCACTGAGTAAAGATAA,mouse iNOS reverse, GGGAGGAGAGGAGAGAGATTTAGTA,mouse GAPDH forward, GCCAAGTATGATGACATCAAGAAGG,and mouse GAPDH reverse, GTGCAGCGAACTTTATTGATGGTAT.

### 4.6. Flow Cytometry

Cell flow cytometry was performed using a NovoCyte Flow Cytometer (ACEA Biosciences, Inc., San Diego, CA, USA) according to the manufacturer’s instructions. Briefly, mAdMSCs (1 × 10^5^) were added to 0.5 mL of perfusion solution (Corning, Manassas, VA, USA). Antibodies (1/100 dilution) were added to the cell suspensions, which were then incubated on ice for 30 min. Fluorescence-activated cell sorting was performed after washing the cells with Brilliant Stain Buffer (BD Biosciences, Franklin Lakes, NJ, USA). The primary antibodies were as follows: Brilliant Violet 421TM Rat Anti-Mouse CD44 (BD Biosciences), Fluorescein Isothiocyanate (FITC) Rat Anti-Mouse CD90.2 (BD Biosciences), PerCP/Cy5.5 Anti-Mouse CD34 (Biolegend, San Diego, CA, USA), and PE/Cy7 Rat Anti-Mouse CD45 (BD Biosciences). Isotype-specific antibodies were used as controls. Flow cytometry was performed as previously described [29,42,43].

### 4.7. Immunofluorescence Analysis

Immunofluorescence analysis was performed according to a published protocol [44]. The antibodies were the same as those described in Section 4.7.

### 4.8. Cell Proliferation

Cells were seeded onto 24-well plates (1.0 × 10^4^ cells/well). On days 1, 3, and 5, the cells were recovered after detachment using trypsin-EDTA, and DNA was extracted using a DNeasy Blood & Tissue Kit (QIAGEN, Hilden, Germany). The final volume of the DNA extraction solution was 100 µL/well.

### 4.9. Cell Differentiation

Adipogenic differentiation was performed using StemXVivo Adipogenic Supplement (CCM011; R&D Systems, Minneapolis, MN, USA), StemXVivo Osteogenic/Adipogenic Base Media, and a Lipid Assay Kit (AK09F; Cosmo Bio Co., Ltd., Tokyo, Japan) according to the manufacturer’s instructions. Osteogenic differentiation was performed using StemXVivo Mouse/Rat Osteogenic Supplement (CCM009; R&D Systems), StemXVivo Osteogenic/Adipogenic Base Media and a Calcified Nodule Staining Kit (AK21, Cosmo Bio Co., Ltd.) according to the manufacturer’s instructions.

### 4.10. LC-MS/MS Analysis of Protein Expression

We used an EzRIPA Lysis Kit (Atto Corporation, Tokyo, Japan) for cell lysis, according to the manufacturer’s instructions. The protein concentrations of the lysates solutions were 3105 µg/mL for C57BL/6 mAdMSCs and 6506 µg/mL for SCID mAdMSCs, and 6.0 µg protein was used for sample preparation, and 0.4 µg protein was analyzed using nanoLC-MS/MS. Nanoflow LC-MS/MS and a database search (Mascot analysis) were conducted by Ikuko Sagawa of the Support Center for Advanced Medical Sciences, Tokushima University Graduate School of Biomedical Sciences. Briefly, proteins were reduced using 10 mM DTT in 8 M urea and Tris buffer with 2 mM EDTA (pH 8.5). Proteins were alkylated using 25 mM iodoacetamide in 8 Murea and Tris buffer with 2 mM EDTA (pH 8.5). The alkylation reactions were subsequently diluted in pig-derived trypsin and incubated at 37 °C overnight. Solid-phase extraction was performed using ZipTip µC18 pipette tips (Merck Millipore, Darmstadt, Germany) to concentrate the peptides. Nano LC-MS/MS was performed using an UltiMate 3000 RSLC Nano System (Thermo Fisher Scientific, Tokyo, Japan) The reconstituted peptides were fractionated using an Acclaim PepMap C18 Trap C18 column (75 µm × 15 cm, 2 µm) (Merck Millipore, Darmstadt, Germany). Solvent A contained 0.1% formic acid and Solvent B contained 80% acetonitrile/0.08% formic acid. Peptides were eluted using a 229-min gradient (4% solvent B in solvent A to 90% solvent B in solvent A), 300 mL/min. The ionization settings of the Orbitrap Elite were as follows: Nanoflow-LC ESI, positive; and capillary voltage, 1.7 kV. Tandem mass spectrometry was performed using Proteome Discoverer software (version 1.4, Thermo Fisher Scientific). We did not perform charge state deconvolution or deisotoping. 

### 4.11. Data Analyses

LC-MS/MS data were quantified according to the theoretical value (emPAI) [45,46,47,48], which was estimated using Scaffold software (Proteome Software, Inc., Portland, OR, USA), and data analyses were performed according to previously reported methods [29,49].

Database search—Mass spectra were extracted using Proteome Discoverer software (version 1.4). We did not perform charge state deconvolution or deisotoping. The analyses of the MS/MS samples were performed using Mascot software (version 2.5.1; Matrix Science, London, UK), which was configured to search the SwissProt_2018_08 database (version unknown, 558125 entries). Mascot was searched using a 0.60-Da fragment-ion mass tolerance and a 5.0-PPM parent ion tolerance. Deamidation of asparagine and glutamine, and oxidation of methionine and carbamidomethyl of cysteine were specified as variable modifications in Mascot.

Protein identification criteria: Proteins were identified using a previously reported method [50]. Briefly, the relative abundance of proteins was estimated by determining the protein abundance index (PAI) and the emPAI. Visualization and validation of the data were performed using Scaffold software to compare samples and identify their biological relevance. We used Scaffold software to validate peptide and protein identifications. Peptide identifications were accepted if it was established with >83.0% probability that they would achieve a false discovery rat (FDR) of <1.0% according to the FDR algorithm of Scaffold. We accepted protein identifications with >95.0% probability that they contained ≥1 identified peptide. Protein probabilities were determined using the Protein Prophet algorithm [51]. Proteins that contained similar peptides, which could not be differentiated based on MS/MS alone, were grouped to satisfy the principles of parsimony. Proteins that shared significant peptide evidence were grouped into clusters. Proteins were annotated with GO terms (goa_uniprot_all.gaf, downloaded 10/14/2016 [26]. Go analysis was performed using the GO analysis function of Scaffold 4 software with data imported from the external GO Annotation Source database (goa_uniprot_all.gaf [downloaded 2016/10/14]) [26].

### 4.12. Statistical Analysis

Statistical analysis was performed using Student’s t-test to compare two samples. Statistical significance was set at *P* < 0.05 for all tests.

## 5. Conclusions

The present study identified proteins differentially expressed by SCID mAdMSCs that may contribute to dysfunctions of the immune system.

## Figures and Tables

**Figure 1 ijms-20-02672-f001:**
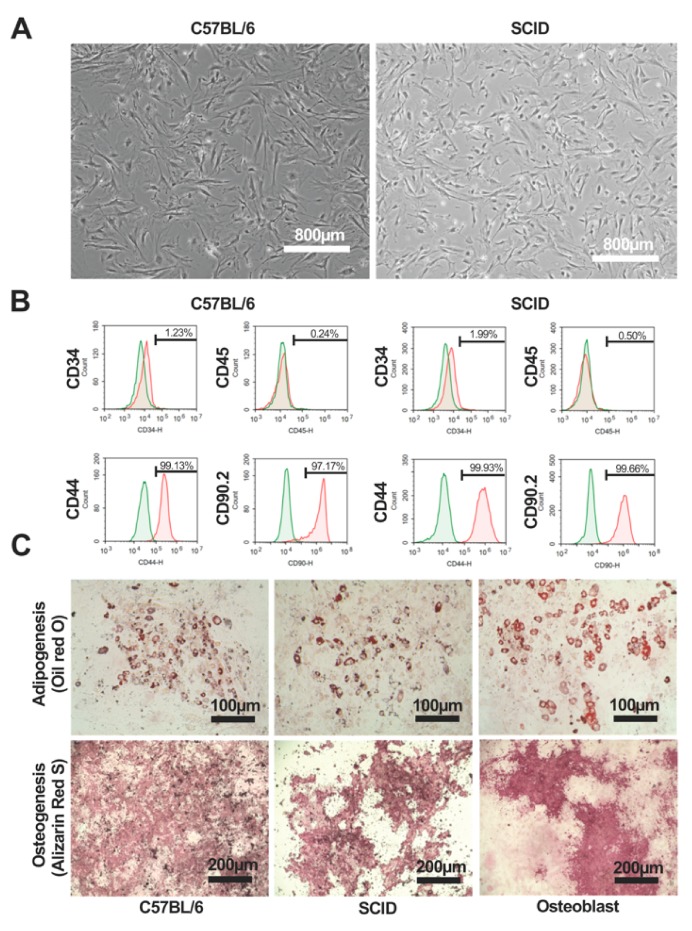
Phenotypes and differentiation potential of C57BL/6 mAdMSCs and SCID mAdMSCs. Morphologies of C57BL/6 mAdMSCs (**A**, left panel) and SCID mAdMSCs (**A**, right panel). Flow cytometric analysis of the expression of cell surface markers of C57BL/6 mAdMSCs (**B**, left panels) and SCID mAdMSCs (**B**, right panel). Representative images of adipocyte (**C**, upper left panel) and osteocyte (**C**, lower left panel) phenotypes of C57BL/6 mAdMSCs cultured in differentiation media. Representative images of adipocyte (**C**, upper middle panel) and osteocyte (**C**, lower middle panel) phenotypes of SCID mAdMSCs cultured in differentiation media. Representative images of adipocyte (**C**, upper right panel) and osteocyte (**C**, lower right panel) phenotypes of osteoblasts cultured in differentiation medium.

**Figure 2 ijms-20-02672-f002:**
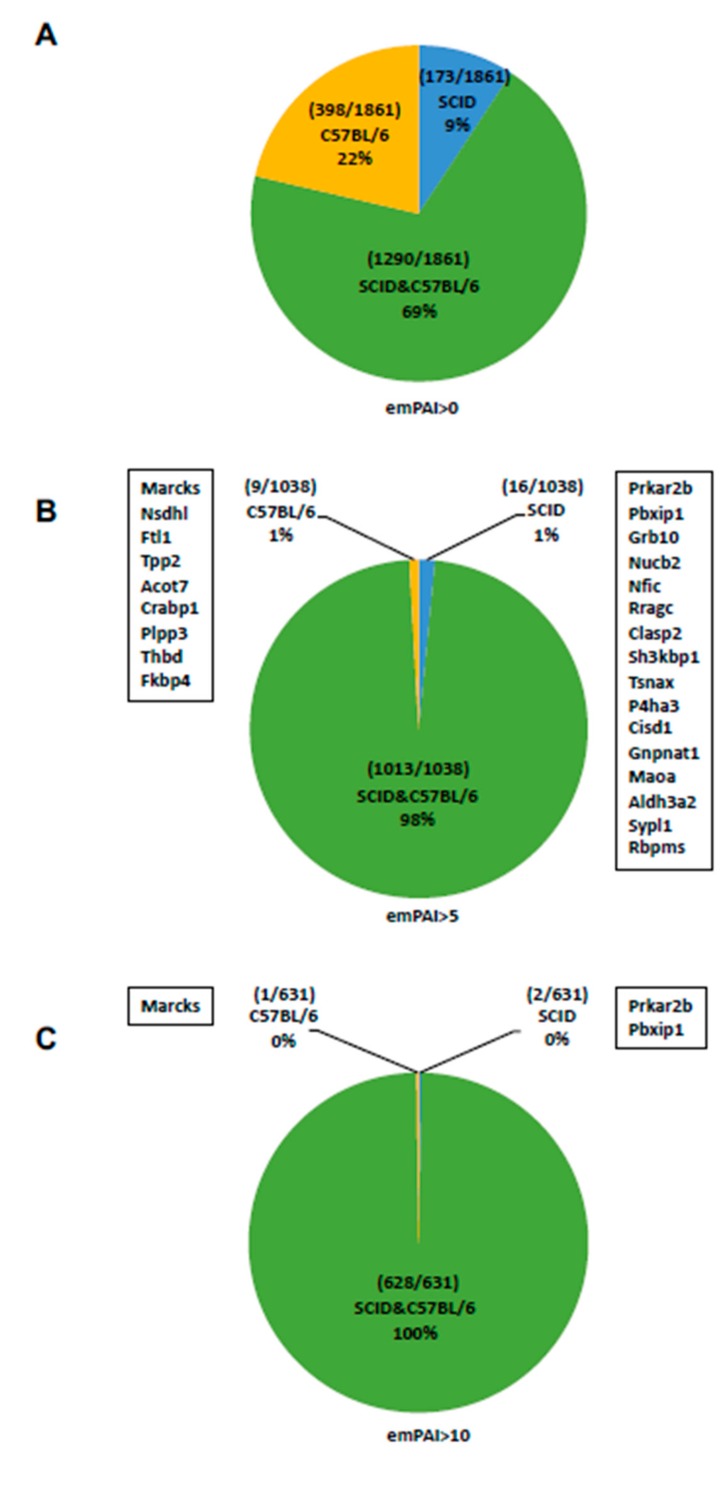
LC-MS/MS analysis of proteins expressed by mADMSCs. emPAI >0: 1688 and 1463 proteins expressed by cultured C57BL/6 mAdMSCs and SCID mAdMSCs, respectively (emPAI >0). Among the 1688 proteins expressed by C57BL/6 mAdMSCs, 398 were unique (emPAI >0). Among the 1463 proteins expressed by SCID mAdMSCs, 173 were unique. The two types of mAdMSCs expressed the same set of 1290 proteins (emPAI >0) (**A**). emPAI >5: C57BL/6 mAdMSCs and SCID mAdMSCs each expressed 1022 and 1029 proteins, respectively, and 9 and 16 proteins were unique to C57BL/6 mAdMSCs and SCID mAdMSCs, respectively. C57BL/6 and SCID mAdMSCs expressed the same set of 1013 proteins (**B**). emPAI >10: Among 629 proteins expressed by C57BL/6 mAdMSCs, 630 were expressed by cultured SCID mAdMSCs, 1 and 2 were uniquely expressed by C57BL/6 mAdMSCs and SCID mAdMSCs, respectively, and the mAdMSCs of both mouse strains expressed the same set of 628 proteins (**C**).

**Figure 3 ijms-20-02672-f003:**
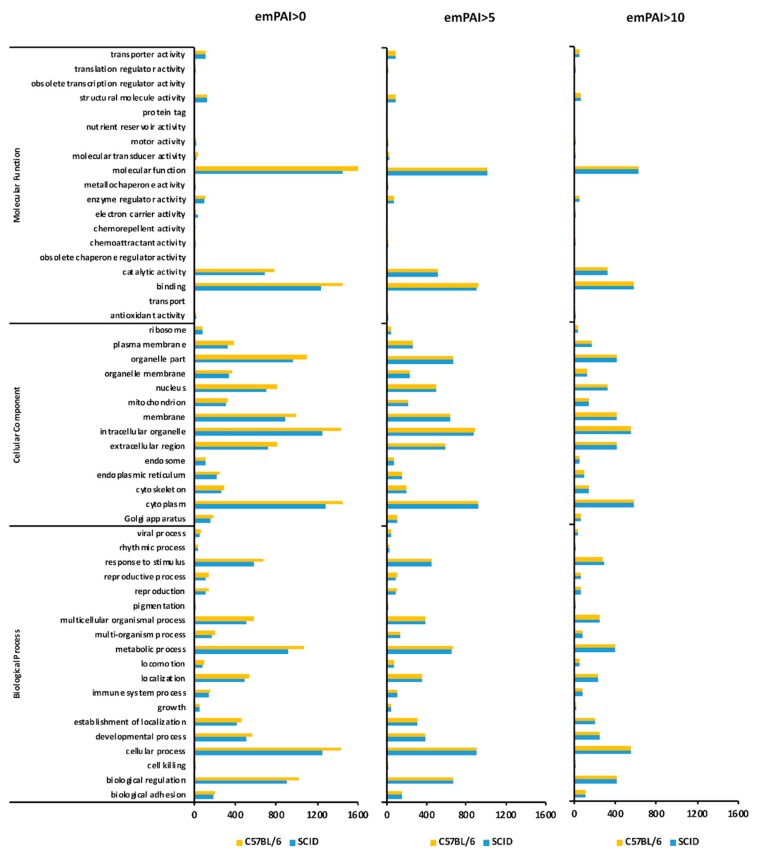
GO analysis of proteins expressed by C57BL/6 mAdMSCs and SCID mAdMSCs. The left, center, and right panels indicate the number of proteins expressed by cultured C57BL/6 and SCID mAdMSCs with emPAI >0, >5, and >10, respectively. The proteins classified in Appendix A are listed according to the GO categories as follows: growth, immune system process, biological regulation, metabolic process, response to stimulus, and viral process. Fifty-seven proteins were classified as growth-related (Appendix A), 165 were classified as immune system process-related (Appendix A), 1115 were classified as biological regulation-related (Appendix A), 1167 were classified as metabolic process-related (Appendix A), 724 were classified as response to stimulus-related (Appendix A), and 62 were classified as viral process-related (Appendix A). The proteins classified in Appendix A are listed according to their molecular function (antioxidant activity, n = 627) (Appendix A). The proteins classified in Appendix A are listed according to their cellular component (membrane). The category “membrane of cellular component-related” was assigned to 1087 proteins (Appendix A). The proteins classified in Appendix A are listed according to their molecular functions (1834 proteins were classified as molecular function-related cultured cell-identified proteins) (Appendix A). The 13 proteins classified in Appendix A are listed according to their classification as B cell-related proteins (Appendix A). The 43 proteins classified in Appendix A are listed according to their categorization as T cell-related proteins. (Appendix A).

**Figure 4 ijms-20-02672-f004:**
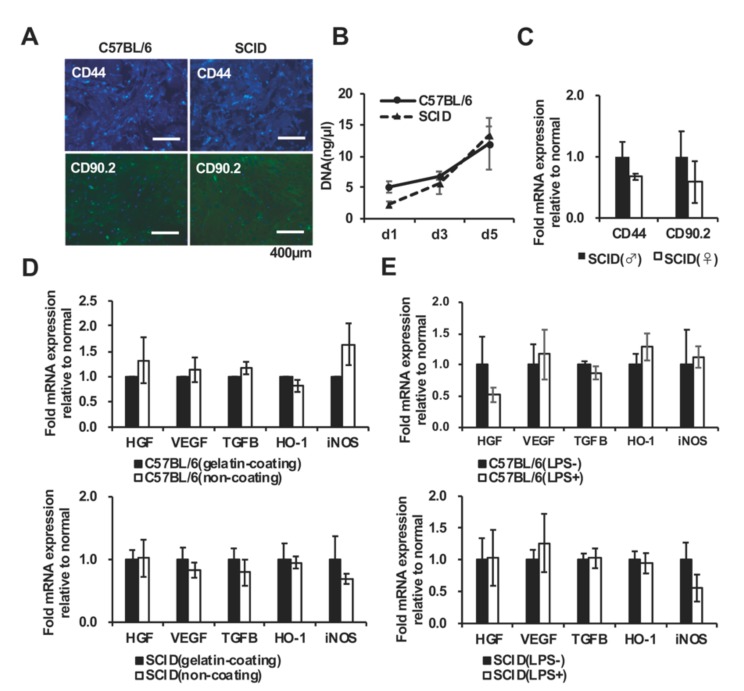
Functions and features of AdMSCs collected from C57BL/6 and SCID mice associated with different culture conditions: The expression of cell surface markers by mAdMSCs (CD44, CD90.2) was evaluated using immunofluorescence with the same antibodies employed for the flow cytometry experiments (Figure 1B) (**A**). The proliferation rates of C57BL/6 mAdMSCs and SCID mAdMSCs were not significantly different on days 3 and 5. The data are expressed as the mean ± SD. The y-axis shows the DNA concentration (µg/µL). Day 1: (C57BL/6 mAdMSCs: 5.08 ± 0.91, SCID mAdMSCs: 2.32 ± 0.46, n = 4). Day 3: (C57BL/6 mAdMSCs: 6.92 ± 0.67, SCID mAdMSCs: 5.74 ± 1.79, n = 4). Day 5: (C57BL/6 mAdMSCs: 12.00 ± 4.13, SCID mAdMSCs: 13.34 ± 1.43, n = 4) (**B**). Real-time PCR analyses (ΔΔCt method) of genes encoding cell surface markers expressed by male SCID mAdMSCs (black bar) and female SCID mAdMSCs (white bar). The expression of the target gene was corrected according to that of HKGs (n = 4) (**C**). Real-time PCR analysis (ΔΔCt method) of the expression of genes encoding growth factors and antioxidant factors by C57BL/6 mAdMSCs (**D**, upper panel) and SCID mAdMSCs (D, lower panel) cultured with (black bar) or without (white bar) a gelatin coating (n = 4 each). Real-time PCR analysis (ΔΔCt method) of the expression of genes encoding growth factors and antioxidant factors expressed by C57BL/6 mAdMSCs cultured in the presence (black bar) or absence (white bar) of LPS (**E**, upper and lower panels, respectively). The levels of the target genes were corrected according to those of HKGs (n = 4) (**E**, upper panel).

**Figure 5 ijms-20-02672-f005:**
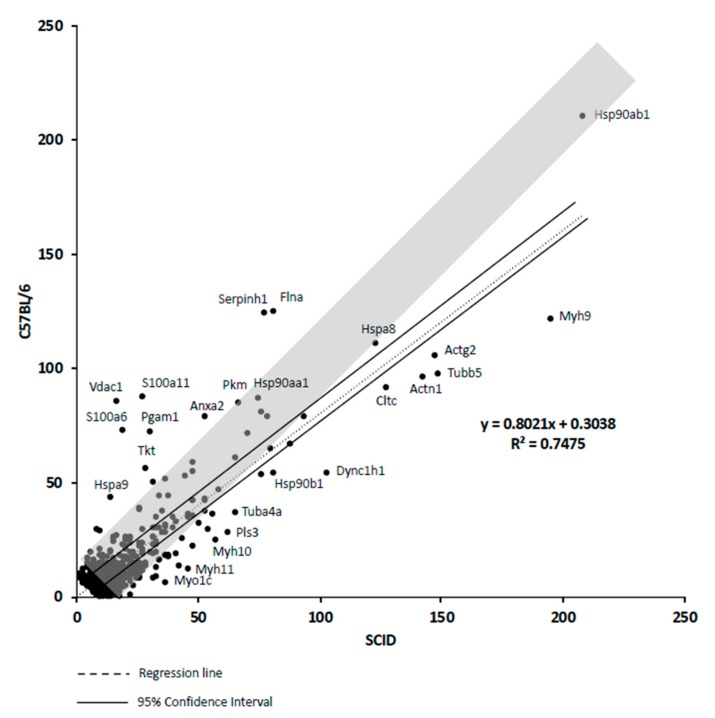
Correlation analysis: A scatter plot showing the correlation (R2 = 0.7475; gray band indicates “R2 = 1”) between the normalized values of proteins expressed by C57BL/6 mAdMSCs and SCID mAdMSCs (n = 622). The parallel lines indicate the 95% confidence interval.

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
