# Peer review of "Identification of Proteins Differentially Expressed by Adipose-derived Mesenchymal Stem Cells Isolated from Immunodeficient Mice"

_ijms, 2019, doi:10.3390/ijms20112672_

Reviewer 1 Report

The manuscript by Yoshiki Nakashima and co-authors was improved significantly.

Unfortunately, the revision track makes difficult to follow the text. 

I suggest the authors revise carefully the manuscript.

Author Response

Response letter

Manuscript ID: ijms-475474

Title: “Immunodeficiency-associated differences observed in mouse adipose-derived mesenchymal stem cells"

Authors: Yoshiki Nakashima, Saifun Nahar, Chika Miyagi-Shiohira, Takao Kinjo, Naoya Kobayashi, Shinji Kitamura, Issei Saitoh, Masami Watanabe, Jiro Fujita and Hirofumi Noguchi.

RESPONSE TO REVIEWER #1:

We apologize for the large number of errors in the previously submitted revised manuscript. We thank the reviewers for giving us the opportunity to revise the text a second time. We clarified the changes in the previously submitted revisions. In addition, the text has been reworked to reflect previous advice that was received from the reviewers. The deletions are clearly indicated in the text. In addition, the sentences that were newly added and corrected in this revision are indicated by yellow highlighting.

Current comment 1:The manuscript by Yoshiki Nakashima and co-authors was improved significantly. Unfortunately, the revision track makes difficult to follow the text. I suggest the authors revise carefully the manuscript.

Response to current comment 1:We carefully reviewed the earlier submitted revisions. Then, we reworked the manuscript carefully according to the reviewer's advice. The text of the re-corrected manuscript is shown below.

Last comment 1:The manuscript by Nakaschima and co-authors describes the differential proteomic analysis in adipose stem cells isolated from C57BL/6 and SCID mice by LC-MS/MS in order to address the immunomodulatory potential of these mesenchymal stem cells. The issue is very important, however, in this current form the manuscript needs to be strongly improved. My opinion is that in the current form the manuscript is not suitable for publication. I encourage the resubmission after a strong revision. Major points:-The introduction must be significantly revised. The aim of the work must be better emphasized.

Response to last comment 1:As suggested, we have now added text about the aim of the work to the Introduction, as follows:

 (Introduction; line 144-164: revision history)

In order to determine whether or not patients suffering from immunodeficiency diseases should use self-collected AdMSCs, we collected AdMSCs from severely immunodeficient mice and examined their cell dysfunction. Severe combined immunodeficiency (SCID) mice are spontaneous mutants that were first identified in C. B-17 strain mice in 1980 by Bosma et al. [18]. The color of these mice is determined by a stereotyped recessive inheritance pattern. Due to the absence of T and B cells, these mice show severe immunodeficiency and are therefore unlikely to reject transplanted heterologous cells or tissues. These mice exhibit severe immunodeficiency because of the absence of T cells and B cells, and are a useful model for studying immunodeficiency caused by HIV infection or radiation exposure. 

In the present study, the cell proliferation, differentiation ability, surface marker expression, and protein expression of mAdMSCs collected SCID mice and normal (c57BL/6) mice were first compared and examined for the presence or absence of cell function abnormalities. In addition, proteins expressed by mAdMSCs were collected from c57BL/6 and SCID mice and analyzed by liquid chromatography (LC-MS/MS) using tandem mass spectrometry. Next, we classified the functions of proteins identified by gene ontology (GO) [19, 20]. The functions of the expressed proteins were examined, focusing on proteins related to the immune response, the viral response, B cells and T cells, and the comparison of the protein expression levels of mAdMSCs obtained from c57BL/6 and SCID mice. The results from the experiments using mAdMSCs collected from these immunodeficient mice were helpful for determining whether or not the function of AdMSCs from patients suffering from immune diseases is a sufficient to allow their use as therapeutic AdMSCs.

Last comment 2: -The results are interesting, however, they must be re-organized. In particular, the comparison between the ASCs from C57BL/6 and SCID mice must be clearly and easier presented. This includes the figures.

Response to last comment 2We have added the immune-related protein expression characteristics of ASCs from C57BL/6 and SCID mice to Supplementary Figure 2B, as suggested. We also added the explanatory text from Supplementary Figure 2B to the Discussion, as follows:

(Discussion; lines 679-688: revision history)

Next[BQ1] , we focused on the expression of proteins related to the viral response (Supplementary Figure 2A) and immune-related proteins (Supplementary Figure 2B) in a GO analysis. The expression levels of viral process-related protein factors (Samhd1) and immune-related protein factors (Ctps1, Hmgb1 and Samhd1) were high in SCID mAdMSCs. The Samhd1 protein protects cells from viral infection and is also involved in the development of cancer and chronic inflammation. Furthermore, the mechanism was clarified recently clarified [40]. In addition, loss-of-function mutations in the Ctps1 gene are associated with the severe and selective impairment of T cell proliferation after antigen stimulation, which shows that Ctps1 is an important checkpoint in adaptive immunity [32]. Hmgb1 is a type of cytokine that is produced in response to injury, infection, and inflammatory stimuli, and is secreted by activated macrophages, mature dendritic cells, and natural killer cells [41].

Last comment 3: -The discussion must be rewritten. The manuscript lacks clear conclusions.

Response to last comment 3As suggested, we have now greatly rewritten the argument and added the following text:

 (Discussion; lines 599-624: revision history)

In the present study, in order to investigate the difference in the protein expression between mAdMSCs from three SCID mice and three normal mice, a comprehensive expression analysis of proteins was performed using LC-MS/MS, which is a proteome analysis method. The ratio of the number of measured peptides to the number of theoretical peptides was linearly related to the logarithm of the protein concentration; with the emPAI defined as the number determined by subtracting 1 from the index of the peptide number ratio. The larger the emPAI value, the greater the amount of proteins. In the proteome analysis by LC-MS/MS, we recognized amino acid codes obtained by digesting proteins in samples that contained peptide fragments and amino acid chains using the shotgun technique. The detected amino acid sequence was identified in the online protein database. In a polymerase chain reaction (PCR), since cDNA can be infinitely amplified artificially, a highly accurate gene expression analysis (e.g., an mRNA expression analysis) is possible even with a trace amount of sample. However, in a protein expression analysis, proteins can be differentiated into amino acid chains, but since the amino acid sequences themselves cannot be amplified, the detection accuracy is limited. Errors in the results of the protein analysis can be attributed to the limits of detection due to the measurement principle and changes in the protein structure due to genetic variations, such as SNPs. When examining a single type of protein, the data reliability was considered to be higher when many peptide sequences were recognized. In this study, we combined inguinal pad fat harvested from three C57BL/6 mice to produce one C57BL/6 mAdMSC culture. We also combined inguinal pad fat harvested from three SCID mice to produce one SCID mAdMSC culture. The reason for this approach is that unlike human beings, the amount of adipose tissue that can be extracted in mice is very small. As a result, both C57BL/6 mAdMSCs and SCID mAdMSCs of these “n = 1” experiments each reflect the properties of cells derived from 3 mice. We recently published a paper on the correlation between different emPAI values (>0, >1, >2, >3, >5, >10) and the results of protein expression analyses. The results showed that, for an emPAI of >10, the presence of protein can be detected with high probability, even if the number of samples is n = 1 [21]. Thus, we considered that the data on proteins with a high emPAI (emPAI > 10) was reliable. 

(Discussion; lines 715-718: revision history)

However, no studies have described the roles that these proteins might play in the therapeutic effect of AdMSCs. The results of this study provided important clues that may help guide the selection of targets when investigating the underlying mechanisms of immune response regulation in AdMSCs. Further, studies on this topic will be needed in the future.

Reviewer 2 Report

In this study, Nakashima et al. comapared the protein expression of adipose-derived MSCs between wild type mice and SCID mice using LC-MS/MS.. They demonstrated that the expression levels of almost proteins are similar in both animal-derived MSCs. Overall, this study is very interesting. However, some corrections are necessary.

1. `ADSCs` is not a general abbreviation of adipose-derived MSCs. `AdMSCs` or `ADMSCs` is proper. ‘ADSCs’ is adipose-derived stem cells.

2. Various corrections of text disappeared during conversion into PDF files. The authors should submit post-corrected text.

3. In Abstract and Discussion sections, final conclusion is not clear.  The authors should refer to whether adipose-derived MSCs in SCID mice have normal immunomodulatory potentials.

Author Response

Response letter

Manuscript ID: ijms-475474

Title: “Immunodeficiency-associated differences observed in mouse adipose-derived mesenchymal stem cells"

Authors: Yoshiki Nakashima, Saifun Nahar, Chika Miyagi-Shiohira, Takao Kinjo, Naoya Kobayashi, Shinji Kitamura, Issei Saitoh, Masami Watanabe, Jiro Fujita and Hirofumi Noguchi.

RESPONSE TO REVIEWER #2:

We apologize for the large number of errors in the previously submitted revised manuscript. We are very grateful to the reviewers for their valuable advice. The reviewer's advice was important for the revision and was useful for helping to convey the content of this paper to the readers in an easy-to-understand manner. We have revised this paper according to the instructions of the reviewer. The deleted text has been clearly indicated. In addition, the sentences that were newly added and corrected in this revision are indicated by yellow highlighting.

Comment 1:In this study, Nakashima et al. comapared the protein expression of adiposederived MSCs between wild type mice and SCID mice using LC-MS/MS.. They demonstrated that the expression levels of almost proteins are similar in both animal-derived MSCs. Overall, this study is very interesting. However, some corrections are necessary. `ADSCs` is not a general abbreviation of adipose-derived MSCs. `AdMSCs` or `ADMSCs` is proper. ‘ADSCs’ is adipose-derived stem cells.

Response 1: We apologize for this issue. All instances of "ADSCs" were corrected to "AdMSCs".

Comment 2: Various corrections of text disappeared during conversion into PDF files. The authors should submit post-corrected text.

Response 2:We apologize for the issues regarding the previously submitted revisions. The version that we are submitting has been re-revised.

Comment 3: In Abstract and Discussion sections, final conclusion is not clear. The authors should refer to whether adipose-derived MSCs in SCID mice have normal immunomodulatory potentials.

Response 3:We consider the reviewer's advice to be very important.  We think it is necessary to clearly show the reader what the conclusion of our research shows. We have deleted the following text from the Abstract, as suggested by the reviewer: 

 (Abstract)

Cell therapy using adipose-derived mesenchymal stem cells (AdMSCs) regulates patient immunity. In this study, we examined the protein expression abnormalities of AdMSCs collected from immunodeficient mice. An analysis of the protein expression of AdMSCs collected from normal mice (c57BL/6) and immunodeficient mice (lacking B and T cells [severe combined immunodeficiency; SCID]) using liquid chromatography with tandem mass spectrometry was carried out. The homology between both groups of protein expression events was 98% (emPAI > 5), or nearly 100% (emPAI > 10). Both groups of AdMSCs expressed cell surface markers (CD44, CD90.2) to the same extent, and there were no abnormalities in the differentiation ability of adipocytes and osteocytes. Furthermore, both groups of cells had the same proliferative capacity. The mRNA expression level of the major expressed proteins of AdMSCs (HGF, VEGF, and TGF-β); or and the expression level of antioxidant- related proteins, such as HO-1 and iNOS, were similar regardless of the presence of a gelatin coat and the induction of inflammation by LPS administration in both groups of cells. However, the expression level of low-to-medium- expression proteins (5 < emPAI<20) involved in immunity was high in AdMSCs of immunodeficient mice. These results indicate that AdMSCs of immunodeficient mice have almost no protein expression dysfunction of protein expression.

As suggested, we have now replacedthe following passages to clarify the conclusions of the Abstract (Short Abstract) and Discussion:

 (Abstract; line 38-55: revision history)

Although cell therapy using adipose-derived mesenchymal stem cells (AdMSCs) regulates patient immunity, the degree to which their quality and function are affected by the individual differences in donor immunodeficiency remains unknown. We subjected mouse AdMSCs (mAdMSCs) collected from 2 research mice: normal mice (c57BL/6) and immunodeficient mice lacking B and T cells (severe combined immunodeficiency mice; SCID mice) to tandem mass spectrometry (LC-MS/MS) liquid chromatography to analyze their protein expression. The protein expression showed 98–100% agreement. These results indicate that the expression of major proteins associated with the therapeutic effect of mAdMSCs was highly similar between the 2 research mice. Furthermore, the expression levels of cell surface markers analyzed by flow cytometry and LC-MS/MS (CD44, CD90.2) were comparable. However, Myh9, Actn1, Canx, Gpi, Tpm1, Eprs, Itgb1, Anxa3, Cnn2, Mapk1, Psme2, Ctps1, Otub1, Psmb6, Hmgb1, Rps19, Sec61a1, Ctnnb1, Glo1, Rpl22, Psma2, Syncrip, Prdx3, Samhd1, Tcaf2, Mapk3, Rps24 and Myo1e, which are associated with the immune system process, were more highly expressed in the SCID mAdMSCs than in the C57BL/6 mAdMSCs. In contrast, Anxa9, Pcbp2, Lgals3, Ppp1r14b, and Psma6, which are also associated with the immune system process, were more highly expressed in the C57BL/6 mAdMSCs than in the SCID mAdMSCs. In this way, we determined the proteins associated with the immune system process that were affected and unaffected by research mouse immunodeficiency, as well as individual differences.

(Discussion; line 700-714: revision history)

In this study, the proteins expressed in MSCs obtained from immunodeficient mice were almost identical to those obtained from normal mice. In this study, the protein expression levels of various functions were compared between the proteins expressed in mAdMSCs generated from three SCID mice and the proteins expressed in mAdMSCs generated from three C57BL/6 mice. The expression levels of representative growth factors and cell surface markers were comparable between SCID and C57BL/6 mAdMSCs. In addition, the expression of 28 viral response-related factors classified by GO classification and the expression levels of 70 immune-related factors were confirmed in both SCID mAdMSCs and C57BL/6 mAdMSCs. However, Myh9, Actn1, Canx, Gpi, Tpm1, Eprs, Itgb1, Anxa3, Cnn2, Mapk1, Psme2, Ctps1, Otub1, Psmb6, Hmgb1, Rps19, Sec61a1, Ctnnb1, Glo1, Rpl22, Psma2, Syncrip, Prdx3, Samhd1, Tcaf2, Mapk3, Rps24, and Myo1e, which are associated with the immune system process, were more highly expressed in SCID mAdMSCs than in C57BL/6 mAdMSCs. In contrast, Anxa9, Pcbp2, Lgals3, Ppp1r14b, and Psma6, which are associated with the immune system process, were more highly expressed in C57BL/6 mAdMSCs than in SCID mAdMSCs. In particular, SCID mAdMSCs expressed a viral process–immune-related protein factor (Samhd1), and immune-related protein factors (Ctps1, Hmgb1, and Samhd1) more strongly than C57BL/6 mAdMSCs

Reviewer 3 Report

The paper submitted by Nakashima and collaborators deals with an appealing topic, however the quality of the presentation is poor and impairs the scientific the soundness of the whole work.

First of all, the text has been submitted in "revision mode", with some lines deleted with a line and replaced in a unclear way that makes it hard to undestand what you are reading.

The abstract and the short abstract come to two different conclusions (any or little dysfunctions?)

The introduction is poor and some references are missing (for example, lines 72-75 need at lest one reference). 

The results are basically not described in this section, but mainly in the figures' captions. Moreover the form of the text hampers the comprehension. Moreover, to me paragraphs 2.2 and 2.3 are the same.

The discussion probably is the section is written in a better way with respect to the other sections, but it is hard to evaluate its significance since the results are so hard to interpret.

The materials and methods section can be improved and some methods could be described more in detail.

In conclusion, it is my opinion that the paper is unsuitable for publication.

Author Response

Response letter

Manuscript ID: ijms-475474

Title: “Immunodeficiency-associated differences observed in mouse adipose-derived mesenchymal stem cells"

Authors: Yoshiki Nakashima, Saifun Nahar, Chika Miyagi-Shiohira, Takao Kinjo, Naoya Kobayashi, Shinji Kitamura, Issei Saitoh, Masami Watanabe, Jiro Fujita and Hirofumi Noguchi.

RESPONSE TO REVIEWER #3:

We apologize for the unclear revision history of the text that we submitted last time. We are very grateful to the reviewers for their advice, which has helped us to improve the quality of this paper. The reviewers pointed out some aspects of our paper that should be improved from a scientific point of view. These points raised by the reviewer have enhanced the academic quality of our paper. The text that has been deleted has been clearly indicated in the text. In addition, sentences that were newly added and corrected in this revision are indicated by yellow highlighting.

Comment 1:The paper submitted by Nakashima and collaborators deals with an appealing topic, however the quality of the presentation is poor and impairs the scientific the soundness of the whole work. First of all, the text has been submitted in "revision mode", with some lines deleted with a line and replaced in a unclear way that makes it hard to undestand what you are reading.

Response 1: We apologize that our previous revision was insufficient. In this revision, the changes that were made in the previous revision are clearly indicated in the text. We also thank our reviewers for their accurate judgment on this paper. Our study data are a comparison of the proteins expressed by fat-derived MSCs produced from fat collected from 3 normal mice and fat-derived MSCs generated from fat collected from 3 SCID mice. We agree that the over-discussion was an attempt to provide a scientific basis to support the pathogenesis of immune deficiency based on data from experiments that used three mice each. Thus, in this revision, the text was revised as a whole to limit the contents to discussion on “individual differences between the expressed proteins of normal mice and immunodeficient mice.”

Comment 2:The abstract and the short abstract come to two different conclusions (any or

little dysfunctions?)

Response 2:We are very grateful to the reviewers for their opinions. We revised the Abstract and Short Abstract to increase their commonality and avoid misleading the reader. 

The following text was deleted from the Abstract, as suggested by the reviewer: 

 (Abstract)

Cell therapy using adipose-derived mesenchymal stem cells (AdMSCs) regulates patient immunity. In this study, we examined the protein expression abnormalities of AdMSCs collected from immunodeficient mice. An analysis of the protein expression of AdMSCs collected from normal mice (c57BL/6) and immunodeficient mice (lacking B and T cells [severe combined immunodeficiency; SCID]) using liquid chromatography with tandem mass spectrometry was carried out. The homology between both groups of protein expression events was 98% (emPAI > 5), or nearly 100% (emPAI > 10). Both groups of AdMSCs expressed cell surface markers (CD44, CD90.2) to the same extent, and there were no abnormalities in the differentiation ability of adipocytes and osteocytes. Furthermore, both groups of cells had the same proliferative capacity. The mRNA expression level of the major expressed proteins of AdMSCs (HGF, VEGF, and TGF-β); or and the expression level of antioxidant- related proteins, such as HO-1 and iNOS, were similar regardless of the presence of a gelatin coat and the induction of inflammation by LPS administration in both groups of cells. However, the expression level of low-to-medium- expression proteins (5 < emPAI<20) involved in immunity was high in AdMSCs of immunodeficient mice. These results indicate that AdMSCs of immunodeficient mice have almost no protein expression dysfunction of protein expression.

As suggested, we have now clarified the conclusion of the Abstract, as follows:

(Abstract; line 38-55: revision history)

Although cell therapy using adipose-derived mesenchymal stem cells (AdMSCs) regulates patient immunity, the degree to which their quality and function are affected by the individual differences in donor immunodeficiency remains unknown. We subjected mouse AdMSCs (mAdMSCs) collected from 2 research mice: normal mice (c57BL/6) and immunodeficient mice lacking B and T cells (severe combined immunodeficiency mice; SCID mice) to tandem mass spectrometry (LC-MS/MS) liquid chromatography to analyze their protein expression. The protein expression showed 98–100% agreement. These results indicate that the expression of major proteins associated with the therapeutic effect of mAdMSCs was highly similar between the 2 research mice. Furthermore, the expression levels of cell surface markers analyzed by flow cytometry and LC-MS/MS (CD44, CD90.2) were comparable. However, Myh9, Actn1, Canx, Gpi, Tpm1, Eprs, Itgb1, Anxa3, Cnn2, Mapk1, Psme2, Ctps1, Otub1, Psmb6, Hmgb1, Rps19, Sec61a1, Ctnnb1, Glo1, Rpl22, Psma2, Syncrip, Prdx3, Samhd1, Tcaf2, Mapk3, Rps24 and Myo1e, which are associated with the immune system process, were more highly expressed in the SCID mAdMSCs than in the C57BL/6 mAdMSCs. In contrast, Anxa9, Pcbp2, Lgals3, Ppp1r14b, and Psma6, which are also associated with the immune system process, were more highly expressed in the C57BL/6 mAdMSCs than in the SCID mAdMSCs. In this way, we determined the proteins associated with the immune system process that were affected and unaffected by research mouse immunodeficiency, as well as individual differences.

We deleted the following text from the Short Abstract, as suggested by the reviewer: 

(Short Abstract; line 59-66: revision history)

We analyzed the protein expression of AdMSCs collected from normal mice (c57BL/6) and immunodeficient mice (severe combined immunodeficiency; [SCID]) were performed. The effect of cell therapy is affected by the donor's immunodeficiency. A protein expression analysis of adipose-derived mesenchymal stem cells (ADSCs) collected from normal mice (c57BL/6) and immunodeficient mice (lacking B and T cells [severe combined immunodeficiency; SCID]) using liquid chromatography with tandem mass spectrometry was conducted. In the immunodeficient group, the expression of mitogen-activated protein kinase 1 and cytidine nucleotide triphosphate synthase 1 proteins related to both B and T cells was increased. These results indicate that ADSCs are cells that constantly express proteins important for immune response control, even in a culture environment.

As suggested, we have now clarified the conclusion of the Short Abstract, as follows:

 (Abstract; line 68-74: revision history)

We analyzed the protein expression of AdMSCs collected from normal mice (c57BL/6) and immunodeficient mice (severe combined immunodeficiency; [SCID]). The protein expression showed 98–100% agreement. However, Myh9, Actn1, Canx, Gpi, Tpm1, Eprs, Itgb1, Anxa3, Cnn2, Mapk1, Psme2, Ctps1, Otub1, Psmb6, Hmgb1, Rps19, Sec61a1, Ctnnb1, Glo1, Rpl22, Psma2, Syncrip, Prdx3, Samhd1, Tcaf2, Mapk3, Rps24, and Myo1e, which are associated with the immune system process, were more highly expressed in the SCID mAdMSCs than in the C57BL/6 mAdMSCs. In contrast, Anxa9, Pcbp2, Lgals3, Ppp1r14b, and Psma6, which are also associated with the immune system process, were more highly expressed in the C57BL/6 mAdMSCs than in SCID mAdMSCs.

Comment 3:The introduction is poor and some references are missing (for example, lines 72-75 need at lest one reference).

Response 3:We appreciate the reviewer's advice. Eight references (17, 18, 19, 20, 21, 23, 25, 26) were added to the Introduction.

Comment 4:The results are basically not described in this section, but mainly in the figures' captions. Moreover the form of the text hampers the comprehension. Moreover, to me paragraphs 2.2 and 2.3 are the same.

Response 4:We apologize for our carelessness. As the reviewer correctly pointed out, the sentences of Results 2.2 and 2.3 overlapped. We therefore deleted Result 2.3.

Comment 5: The discussion probably is the section is written in a better way with respect to

the other sections, but it is hard to evaluate its significance since the results

are so hard to interpret.

Response 5:We appreciate the important comment from the reviewer with regard to the readability of the text. Based on the reviewer's comments, the Discussion was divided into several sections. Furthermore, the following sentence was added to the Discussion.

Based on the reviewer's suggestion, we added the following text to the Discussion:

(Discussion; lines 597: revision history)

•The reliability of the comparison of protein expression data obtained by LC-MS/MS in the present study

 (Discussion; lines 599-623: revision history)

In the present study, in order to investigate the difference in the protein expression between mAdMSCs from three SCID mice and three normal mice, a comprehensive expression analysis of proteins was performed using LC-MS/MS, which is a proteome analysis method. The ratio of the number of measured peptides to the number of theoretical peptides was linearly related to the logarithm of the protein concentration; with the emPAI defined as the number determined by subtracting 1 from the index of the peptide number ratio. The larger the emPAI value, the greater the amount of proteins. In the proteome analysis by LC-MS/MS, we recognized amino acid codes obtained by digesting proteins in samples that contained peptide fragments and amino acid chains using the shotgun technique. The detected amino acid sequence was identified in the online protein database. In a polymerase chain reaction (PCR), since cDNA can be infinitely amplified artificially, a highly accurate gene expression analysis (e.g., an mRNA expression analysis) is possible even with a trace amount of sample. However, in a protein expression analysis, proteins can be differentiated into amino acid chains, but since the amino acid sequences themselves cannot be amplified, the detection accuracy is limited. Errors in the results of the protein analysis can be attributed to the limits of detection due to the measurement principle and changes in the protein structure due to genetic variations, such as SNPs. When examining a single type of protein, the data reliability was considered to be higher when many peptide sequences were recognized. In this study, we combined inguinal pad fat harvested from three C57BL/6 mice to produce one C57BL/6 mAdMSC culture. We also combined inguinal pad fat harvested from three SCID mice to produce one SCID mAdMSC culture. The reason for this approach is that unlike human beings, the amount of adipose tissue that can be extracted in mice is very small. As a result, both C57BL/6 mAdMSCs and SCID mAdMSCs of these “n = 1” experiments each reflect the properties of cells derived from 3 mice. We recently published a paper on the correlation between different emPAI values (>0, >1, >2, >3, >5, >10) and the results of protein expression analyses. The results showed that, for an emPAI of >10, the presence of protein can be detected with high probability, even if the number of samples is n = 1 [21].

 (Discussion; lines 624-627: revision history)

However, the data of these “n = 1” experiments was strongly affected by individual differences. Thus, the data of the protein expression analyses using LC-MS/MS in this study revealed that the individual differences of SCID mice (with immunodeficiency) were in line with those of normal mice.

(Discussion; lines 629: revision history)

•The expression of T cell/B cell-related proteins, as determined by LC-MS/MS.

 (Discussion; lines 653-654: revision history)

• The protein expression analysis using LC-MS/MS demonstrates the total protein expression of C57BL/6 mAdMSCs and SCID mAdMSCs.

 (Discussion; lines 665: revision history)

• Examination of the immune process.

 (Discussion; lines 677: revision history)

• Examination about viral response.

 (Discussion; lines 690: revision history)

• Overall consideration of the results of the present study.

Comment 6: The materials and methods section can be improved and some methods could

be described more in detail.

Response 6: We thank the reviewers for pointing out. Added a detailed explanation to "Materials and Methods" according to the instructions of the reviewer.

In accordance with the reviewer's suggestion, we added the following text to the Materials and Methods:

 (Materials and Methods"; lines 1791-801: revision history)

Cell flow cytometry was performed using a NovoCyte®Flow Cytometer (ACEA Biosciences, Inc., San Diego, CA, USA), according to the manufacturer's instructions. Briefly, mAdMSCs (1 ×105cells) were added to 0.5 mL of perfusion solution (Corning, Manassas, VA, USA). Each of the antibodies (1/100 of the volume) was added to the cell admixture. The admixture was then incubated on ice for 30 min. Fluorescence-activated cell sorting was performed after washing the cells with Brilliant Stain Buffer (BD Biosciences, Franklin Lakes, NJ, USA), and fluorescence-activated cell sorting (FACS) measurements were conducted. The following primary antibodies were used: Brilliant Violet 421TM Rat Anti-Mouse CD44 (BD Biosciences), Fluorescein Isothiocyanate (FITC) Rat Anti-Mouse CD90.2 (BD Biosciences), PerCP/Cy5.5 Anti-Mouse CD34 (Biolegend, San Diego, CA, USA), and PE/Cy7 Rat Anti-Mouse CD45 (BD Biosciences). Isotype-identical antibodies were used as controls.

(Materials and Methods; lines 827-840: revision history)

Briefly, protein-containing solutions were reduced with 10 mM DTT/8 M urea and Tris buffer containing 2 mM EDTA (pH 8.5), alkylated with 25 mM iodoacetamide/8 M Urea and Tris buffer containing 2 mM EDTA (pH 8.5), which was subsequently diluted with trypsin (pig-derived trypsin) and digested overnight at 37°C. Solid-phase extraction (SPE) in ZipTip μC18 pipette tips (Merck Millipore, Darmstadt, Germany) was performed to concentrate the peptides. An UltiMate 3000 RSLC nano system (Thermo Fisher Scientific) was used to perform nano LC-MS/MS. The reconstituted peptides were injected into an Acclaim PepMap C18 trap column (75 μm × 15 cm, 2 μm, C18) (Merck Millipore, Darmstadt, Germany). Solvent A was 0.1% formic acid. Solvent B was 80% acetonitrile/0.08% formic acid. Peptides were eluted in a 229-min gradient (4% solvent B in solvent A to 90% solvent B in solvent A) at 300 nl/min. The ionization settings of the Orbitrap Elite were as follows: Nanoflow-LC ESI, positive; and capillary voltage, 1.7 kV. Tandem mass spectrometry was performed with the Proteome Discoverer software program (version 1.4, Thermo Fisher Scientific). We did not perform charge stated deconvolution or deisotoping.

(Materials and Methods; lines 843, 848-873: revision history)

4.11. Data analyses

DATABASE SEARCH—Tandem mass spectra were extracted using the Proteome Discoverer software program (version 1.4). We did not perform charge state deconvolution or deisotoping. The analyses of the MS/MS samples were all performed using the Mascot software program (version 2.5.1; Matrix Science, London, UK). Mascot was set up to search the SwissProt_2018_08 database (version unknown, 558125 entries). Mascot was searched with a 0.60-Da fragment ion mass tolerance and a 5.0-PPM parent ion tolerance. Deamidation of asparagine and glutamine, and oxidation of methionine and carbamidomethyl of cysteine were specified as variable modifications in Mascot.

CRITERIA FOR PROTEIN IDENTIFICATION—A comprehensive expression analysis of proteins was performed by LC-MS/MS, according a previously reported method [52]. In brief, the relative abundance of proteins identified by LC-MS/MS was estimated by determining both the protein abundance index (PAI) and the exponentially modified protein abundance index (emPAI). The visualized and validated complex LC-MS/MS proteomics experiments were performed using the Scaffold software program (version 4.8.7, Proteome Software Inc., Portland, OR, USA) (http://www.proteomesoftware.com/) to compare samples and identify biological relevance. We used the Scaffold software program (version 4.8.7, Proteome Software Inc., Portland, OR) to validate MS/MS-based peptide and protein identifications. Peptide identifications were accepted if it was established with >93.0% probability that they would achieve an FDR of<1.0% by="" the="" local="" fdr="" algorithm="" of="" scaffold.="" protein="" identifications="" were="" accepted="" if="" it="" was="" established="" with="">95.0% probability that they contained at least 1 identified peptide. The protein probabilities were determined by the Protein Prophet algorithm [53]. The proteins that contained similar peptides, which could not be differentiated based on MS/MS alone were grouped to satisfy the principles of parsimony. Proteins that shared significant peptide evidence were grouped into clusters. Proteins were annotated with GO terms from goa_uniprot_all.gaf (downloaded 2016/10/14) [27].

Comment 7: In conclusion, it is my opinion that the paper is unsuitable for publication.

Response 7:In this study, we detected the mAdMSC-expressed proteins associated with the immune system and viral processes that were affected and unaffected by the immunodeficiency of the model mice, as well as individual differences. As suggested by the reviewer, we have now deleted the following text from the Conclusions:

 (Conclusions)

The expression of Severe immunodeficiency mice (SCID) mAdMSC-expressed growth factors, cell surface markers, virus response-related factors, and expression of major factors related to immune-related factors in SCID mAdMSCs are not abnormally compared to markedly different from those in C57BL/6 mAdMSCs. However, SCID mAdMSCs was showed a stronger expression of certain viral process-related protein factors (Samhd1) and immune-related protein factors (Ctps1, Hmgb1 and Samhd1) expressed more strongly than C57BL/6 mAdMSCs. In this way, we determined the proteins associated with the immune system process that were affected and unaffected by research mice immunodeficiency as well as individual differences.

As suggested, we have now replacedthe text of the Conclusions, as follows:

 (Conclusions; line 875-877: revision history)

In this study, we detected the mAdMSC-expressed proteins associated with the immune system process and viral process that were affected and unaffected by immunodeficiency or individual differences in research mice.

Last comment 1:  In this manuscript, Nakashima et al compared the protein expression profile of adipose tissue mesenchymal stem cells they isolated from immunocompetent and immunodeficient mice. With an unclear rationale, the authors concluded that A-MSC from both mice constantly express proteins linked to immune response control at least in vitro. This descriptive and comparative study looks very preliminary and its rationale should be more elaborated to help the readers easily following the results. Other experiments are needed to strengthen the quality of the study like the use of additional male samples, the comparison of specific features of the cells, and their immunomodulatory properties. I also strongly advice the authors top rewrite the abstract, the results and the discussion sections.

- How many mice were used in that study?

Response to last comment 1:As suggested, we compared the mRNA expression of cell surface markers (CD44, CD90.2) expressed in male and female SCID mice in Figure 4C in order to present data on AdMSCs collected from both male and female mice. To compare the immunomodulatory properties of AdMSCs harvested from normal and SCID mice, Figure 4E includes new data on the major protein mRNA expression under pro-inflammatory conditions. Furthermore, data on the expression of virus response-related proteins have now been added to Supplementary Figure 2A, and similar data on the expression of immunomodulatory proteins have been added to Supplementary Figure 2B. In addition, we substantially revised the Abstract, Results and Discussion, and the number of mice used for the study has been mentioned in the Materials & Methods.

In accordance with the reviewer’s suggestion, we deleted the following sentences:

 (Abstract; line 26-38: revision history)

Cell therapy using adipose-derived mesenchymal stem cells (ADSCs) regulates patient immunity. However, may aspects of the mechanism are unclear, particularly how cell therapy is affected by the donor's immunodeficiency. An analysis of the protein expression of ADSCs collected from normal mice (c57BL/6) and immunodeficient mice (lacking B and T cells [severe combined immunodeficiency; SCID]) using liquid chromatography with tandem mass spectrometry was carried out. The protein expression events were nearly 100% consistent between both groups. Mitogen-activated protein kinase 1 (Mapk1) and cytidine nucleotide triphosphate synthase 1 (Ctps1) associated with B cells were highly expressed in the mADSC (SCID) group, as were myosin IIA heavy chain protein (Myh9), Mapk1 and Ctps1 associated with T cells. The expression of Mapk1 and Ctps1 proteins related to both B and T cells was increased in the mADSC (SCID) group. Ctps1 is an important checkpoint factor in adaptive immunity. These results indicate that ADSCs are cells that constantly express proteins important for immune response control, even in a culture environment.

This deleted text was replaced with the following:

 (Abstract; line 38-55: revision history)

Although cell therapy using adipose-derived mesenchymal stem cells (AdMSCs) regulates patient immunity, the degree to which their quality and function are affected by the individual differences in donor immunodeficiency remains unknown. We subjected mouse AdMSCs (mAdMSCs) collected from 2 research mice: normal mice (c57BL/6) and immunodeficient mice lacking B and T cells (severe combined immunodeficiency mice; SCID mice) to tandem mass spectrometry (LC-MS/MS) liquid chromatography to analyze their protein expression. The protein expression showed 98–100% agreement. These results indicate that the expression of major proteins associated with the therapeutic effect of mAdMSCs was highly similar between the 2 research mice. Furthermore, the expression levels of cell surface markers analyzed by flow cytometry and LC-MS/MS (CD44, CD90.2) were comparable. However, Myh9, Actn1, Canx, Gpi, Tpm1, Eprs, Itgb1, Anxa3, Cnn2, Mapk1, Psme2, Ctps1, Otub1, Psmb6, Hmgb1, Rps19, Sec61a1, Ctnnb1, Glo1, Rpl22, Psma2, Syncrip, Prdx3, Samhd1, Tcaf2, Mapk3, Rps24 and Myo1e, which are associated with the immune system process, were more highly expressed in the SCID mAdMSCs than in the C57BL/6 mAdMSCs. In contrast, Anxa9, Pcbp2, Lgals3, Ppp1r14b, and Psma6, which are also associated with the immune system process, were more highly expressed in the C57BL/6 mAdMSCs than in the SCID mAdMSCs. In this way, we determined the proteins associated with the immune system process that were affected and unaffected by research mouse immunodeficiency, as well as individual differences.

The following sentences were added to the Materials and Methods:

 (Materials and Methods; line 743-744: revision history)

4.3. Isolation of AdMSCsfrom mouse Adipose tissuevia the inguinal pad fat

Adipose tissue was obtained from the inguinal pad fat of three 8-week-old mice (female, n = 3).

Last comment 2: Additional major comments: - Many information on the MSC used are lacking [Passage number, viability level, proliferation potential (cumulative population doubling), initial recovered yield, …]

Response to last comment 2We have now mentioned the MSC passage number in the Materials & Methods, as suggested. The survival and proliferation data were added to Figure 4B. However, we were unable to calculate the initial recovered yield due to contamination by blood cells during the process of separating AdMSCs from mouse adipose tissue.

The following text was added to the Materials and Methods:

 (Materials and Methods)

4.4. Preparation of mAdMSCSs

The mAdMSCs used in this study has been passaged three times.

The following text was added to Figure 4B.

(Figure legend; Figure 4B)

The cellular proliferation rates of C57BL/6 mAdMSCs and SCID mAdMSCs did not differ to a statistically significant extent on days 3 and 5. The data are expressed as the mean ± SD. The vertical axis shows the DNA concentration (μg/μl). Day 1: (C57BL/6 mAdMSCs: 5.08 ± 0.91, SCID mAdMSCs: 2.32 ± 0.46, n = 4). Day 3: (C57BL/6 mAdMSCs: 6.92 ± 0.67, SCID mAdMSCs: 5.74 ± 1.79, n = 4). Day 5: (C57BL/6 mAdMSCs: 12.00 ± 4.13, SCID mAdMSCs: 13.34 ± 1.43, n = 4) (B).

Last comment 3:- How those MSC behave in an inflammatory environment?

Response to last comment 3:As suggested, we added data on the major protein mRNA expression of AdMSCs in an inflammatory environment induced by LPS to Figure 4E. The following text to Figure 4E.

(Figure legend; Figure 4E)

The results of a real-time PCR to detect growth factors and antioxidant factors of LPS-free C57BL/6 mAdMSCs (black bar) and LPS-added C57BL/6 mAdMSCs (white bar). The expression was calculated using the ΔΔCt method. The expression of the target gene was corrected by the expression of the housekeeping gene. The relative values are indicated (n = 4) (E, upper panel). The results of a real-time PCR to detect growth factors and antioxidant factors of LPS-free SCID mAdMSCs (black bar) and LPS-added SCID mAdMSCs (white bar). The expression was calculated using the ΔΔCt method. The expression of the target gene was corrected by the expression of the housekeeping gene. The relative values are indicated (n = 4) (E, lower panel).

Last comment 4:- To support the claimed conclusion related to the effect of in vitro culture on the A-MSC phenotype, the authors should compare the plated A-MSC to nonplated ones.

Response to last comment 4:As suggested, we conducted experiments based on the presence of a gelatin coating, which is the scaffolding material for the plate cell adhesion of AdMSCs. Data on the major protein mRNA expression of AdMSCs collected from normal and SCID mice cultured with and without such a coating are shown in Figure 4D.The following text was added to Figure 4D:

(Figure legend; Figure 4D)

The results of a real-time PCR to detect growth factors and antioxidant factors of gelatin-coated dish-cultured C57BL/6 mAdMSCs (black bar) and non-coated dish-cultured C57BL/6 mAdMSCs (white bar). The expression was calculated using the ΔΔCt method. The expression of the target gene was corrected by the expression of the housekeeping gene. The relative values are indicated (n = 4) (D, upper panel). The results of a real-time PCR to detect growth factors and antioxidant factors of gelatin-coated dish-cultured SCID mAdMSCs (black bar) and non-coated dish-cultured SCID mAdMSCs (white bar). The expression was calculated using the ΔΔCt method. The expression of the target gene was corrected by the expression of the housekeeping gene. The relative values are indicated (n = 4) (D, lower panel). 

Last comment 5:- The main proteins significantly expressed should be reanalyzed by another

experimental approach. The secretome protein levels should also be checked

Response to last comment 5:As suggested, we presented the expression of cell surface markers (CD44, CD90.2) as fluorescence immunomicrograph data in Figure 4A and the mRNA expression data in Figure 4C. In addition, data on the major protein mRNA expression (HGF, VEGF, TGFB, HO-1, iNOS) of AdMSCs were added to Figures 4D and 4E. This mRNA expression was also evaluated using Western blotting.The following text was added to Figure 4A and 4C:

 (Figure legend; Figure 4A)

The expression status of cell surface markers of mAdMSCs (CD44, CD90.2) was evaluated by immunofluorescence staining. The antibodies were the same as those used in the flow cytometry experiments

 (Figure legend; Figure 4C)

The results of a real-time PCR to detect cell surface markers of male SCID mAdMSCs (black bar) and female SCID mAdMSCs (white bar). The expression was calculated using the ΔΔCt method. The expression of the target gene was corrected by the expression of the housekeeping gene. The relative values are indicated (n = 4) (C). (Figure legend; Figure 4D)

Please see Response to last comment 4.

(Figure legend; Figure 4E)

Please see Response to last comment 3.

Last comment 6:Detailed comments:

- The reference list should be revised and more organized. Ex: 3 references for citing dental pulp MSC is too much as the statement is very general, Bosma et al should be inserted, …

Response to last comment 6:As suggested, we revised our references in relation to the collection of MSCs from various sites. In addition, Dr. Bosma's paper was added to the references.

Last comment 7:- Figure 1 should be completely reorganized for a better clarity. A, should be for photos, B, for FACS analysis and C for differentiation of both groups

Response to last comment 7:As suggested by the reviewer, we reorganized Figure 1.

Last comment 8:- Page 4: Why the amount of recovered protein yield of SCID is 2 times higher (6506 μg/ml) than in C57BL/6 (3105 μg/ml)?

Response to last comment 8:The difference in the cell mass of AdMSCs collected for the LC-MS/MS analysis is believed to have influenced this discrepancy. However, since the total amount of protein injected into the measurement device was consistent, the measurement data were not affected.

Last comment 9:- The quality of the photos related to adipogenic differentiation is not convincing.

Response to last comment 9:As suggested, we reexamined the induction of differentiation into adipocytes, and the picture was revised.

The previous method (described below) was been removed.

 (Materials and Methods; line 820: revision history)

4.6. Cell differentiation

Adipogenic and osteogenic differentiation were performed as described previously[41, 44].

This was replaced with the following text:

(Materials and Methods; line 812-819: revision history)

4.9. Cell differentiation

Adipogenic differentiation was performed using StemXVivo Adipogenic Supplement (CCM011; R&D Systems, Minneapolis, MN, USA), StemXVivo Osteogenic/Adipogenic Base Media and a Lipid Assay Kit (AK09F; Cosmo Bio Co., Ltd., Tokyo, Japan) according to the manufacturer's instructions. Osteogenic differentiation was performed using StemXVivo Mouse/Rat Osteogenic Supplement (CCM009; R&D Systems, Minneapolis, MN, USA), StemXVivo Osteogenic/Adipogenic Base Media and a Calcified Nodule Staining Kit (AK21, Cosmo Bio Co., Ltd.), according to the manufacturer's instructions.

Last comment 10:- Please use osteoblasts or Osteocytes!

Response to last comment 10:We reexamined osteocyte differentiation induction using osteoblasts harvested from the mouse parietal bone, as suggested.The following text was added to Figure 4C:

 (Figure legend; Figure 1C)

Representative images of adipocyte (C, upper left panel) and osteocyte differentiation (C, lower left panel) from C57BL/6 mAdMSCs cultured in differentiation medium. Representative images of adipocyte (C, upper middle panel) and osteocyte differentiation (C, lower middle panel) from SCID mAdMSCs cultured in differentiation medium. Representative images of adipocyte (C, upper right panel) and osteocyte differentiation (C, lower right panel) from osteoblasts cultured in differentiation medium.

 (Materials and Methods; line 730-731: revision history)

4.1. Reagents and materials

Mouse osteoblasts from cranial bone were obtained (Code No. OBC12C) from Cosmo Bio Co., Ltd. (Tokyo, Japan).

Last comment 11:The paragraphs and subparagraphs of the results sections should be reorganized

Response to last comment 11:We reorganized the paragraphs and subparagraphs of the Results section, as suggested.

Last comment 12:- Paragraphs 2.10 & 2.11: no results description was provided!

Response to last comment 12:As suggested, we added a description of the results in paragraphs 2.10 and 2.11.

The following text was added to the Results:

 (Results)

2.8. The GO classifications of the proteins expressed in C57BL/6 and SCID mAdMSC samples (emPAI > 0)

The condition of emPAI > 0 identified 60 “viral process-related proteins” for C57BL/6 mAdMSCs and 53 for SCID mAdMSCs. The condition of emPAI > 0 identified 154 “Immune system process-related proteins” for C57BL/6 mAdMSCs and 137 for SCID mAdMSCs.

(Results; line 365-367: revision history)

2.9. The GO classifications of the proteins expressed in C57BL/6 and SCID mAdMSC samples (emPAI > 5)The condition of emPAI > 5 identified “44 viral process-related proteins” for C57BL/6 mAdMSCs and 44 for SCID mAdMSCs. The condition of emPAI > 0 identified 110 “immune system process-related proteins” for C57BL/6 mAdMSCs and 109 for SCID mAdMSCs.

 (Results; line 377-379: revision history)

2.10. The GO classifications of the proteins expressed in C57BL/6 and SCID mAdMSC samples (emPAI > 10)

The condition of emPAI > 10 identified 28 “viral process-related proteins” for C57BL/6 mAdMSCs and 28 for SCID mAdMSCs. The condition of emPAI > 0 identified 75 “immune system process-related proteins” for C57BL/6 mAdMSCs, and 76 for SCID mAdMSCs.

Last comment 13:- The message of the title should be improved and more attractive

Response to last comment 13:We have revised the title according to the reviewer's advice.

The previous title (shown below) has been removed.

(previous title)

Immunodeficiency-associated differences observed in mouse adipose-derived mesenchymal stem cells

This title was replaced with the following title:

 (new title)

Individual differences in mouse adipose-derived mesenchymal stem cells collected from immunodeficient mice

Last comment 14:- Several typo and grammatical errors should be corrected

Response to last comment 14:We have now carefully re-checked the entire paper and corrected these errors. Furthermore, the whole manuscript was rechecked by a professional editor, who is a medical English expert and a native speaker of English.

Round  2

Reviewer 1 Report

The manuscript has been improved.

I suggest changing the title, in order to be more representative of the manuscript.

Author Response

May 25, 2019 Manuscript ID: ijms-475474 Siya Jiang Assistant Editor International Journal of Molecular Science Dear Ms Jiang, Our point-by-point responses to the reviewers’ comments about our revised manuscript # ijms-475474 (revised title: “Identification of Proteins Differentially Expressed by Adipose-derived Mesenchymal Stem Cells Isolated from Immunodeficient Mice”) are as follows: Reviewer #1: Comment: I suggest changing the title, in order to be more representative of the manuscript. Response to Reviewer #1: We revised the title as shown above.

Reviewer 3 Report

Nakashima and collaborators revised their paper in the attempt to answer to reviewers' comments, but in my opinion even in the revised version the paper in still unsuitable for publication.

First of all, I have to say again that the paper is hard to read. I understand that the authors want to explicit their revision work, but in this form the readibility of the text is compromised. Moreover, some figures are misplaced and/or the legends don't match with the figures.

Coming to the scientific content of the paper. it has not been improved at all. The samples analyzed are basically one for each mouse type, even if they are made up by pooling three mice each. At a certain point in the paper the authors make a comparison between cells obtained from male and female donors (honestly, comparing the text and the related figure I can't understad if they compare female donors from each mouse typer or male vs female donors for the same mouse type). Is is not clear how it was possible since the cells were pooled to obtain a single sample for each mouse strain.

Author Response

May 25, 2019 Manuscript ID: ijms-475474 Siya Jiang Assistant Editor International Journal of Molecular Science Dear Ms Jiang, Our point-by-point responses to the reviewers’ comments about our revised manuscript # ijms-475474 (revised title: “Identification of Proteins Differentially Expressed by Adipose-derived Mesenchymal Stem Cells Isolated from Immunodeficient Mice”) are as follows: Reviewer #3: Response to Reviewer #3: We thank the Reviewer for the incisive suggestions that helped to greatly enhance the quality of our manuscript Comment 1: Nakashima and collaborators revised their paper in the attempt to answer to reviewers' comments, but in my opinion even in the revised version the paper in still unsuitable for publication. First of all, I have to say again that the paper is hard to read. I understand that the authors want to explicit their revision work, but in this form the readibility of the text is compromised. Moreover, some figures are misplaced and/or the legends don't match with the figures. Response 1: This manuscript has been carefully reviewed by an experienced medical editor whose first language is English and who specializes in the editing of papers written by physicians and scientists whose native language is not English. Thus, the manuscript has undergone extensive revisions to improve the English and the clarity of our narrative. However, we also consider that the opinion of this reviewer #3 who is an expert in this field has priority. Comment 2: Coming to the scientific content of the paper. it has not been improved at all. The samples analyzed are basically one for each mouse type, even if they are made up by pooling three mice each. At a certain point in the paper the authors make a comparison between cells obtained from male and female donors (honestly, comparing the text and the related figure I can't understad if they compare female donors from each mouse typer or male vs female donors for the same mouse type). Is is not clear how it was possible since the cells were pooled to obtain a single sample for each mouse strain. Response 2: The data contributed by our study were acquired through a comprehensive analysis of proteins differentially expressed by adipose mesenchymal stem cells of SCID mice vs those isolated from an immunocompetent strain. Our findings will enhance efforts to evaluate whether individuals with immune dysfunction can donate adipose-derived MSCs for regenerative medicine. We believe that the substantive revisions to the manuscript as well as improvements to the English sufficiently address this Reviewer’s criticisms. It is important to note that we have added a caveat to the Discussion stating that “This is a pilot study of protein expression, and it should be noted therefore that the reliability of the data is influenced by the research method.”

Round  3

Reviewer 3 Report

I have to say again that the the revision tracks make difficult to follow the text.  

I understand the authors' point of view, but even if the paper has greatly improved, I am not sure about its importance due to the limited sample size.

The authors are aware of the problem (as reported in the discussion), but this limitation in my opinion need to be taken into account by the editor.

As a reviewer, I suggest to accept the paper after minor revision (mainly text editing, or at least a close revision of the final version that is not easy to read in the present revision mode) even if its significance is low.

Author Response

May 28, 2019

Manuscript ID: ijms-475474

Siya Jiang

Assistant Editor 

International Journal of Molecular Science

Dear Ms Jiang,

Our point-by-point responses to the reviewers’ comments about our revised manuscript # ijms-475474 (revised title: “Identification of Proteins Differentially Expressed by Adipose-derived Mesenchymal Stem Cells Isolated from Immunodeficient Mice”) are as follows:

Reviewer #3:

Comment 1: As a reviewer, I suggest to accept the paper after minor revision (mainly text editing, or at least a close revision of the final version that is not easy to read in the present revision mode) even if its significance is low.

Response 1: We revised our manuscript.

We hope that the revised manuscript is suitable for publication in the International Journal of Molecular Science.

Sincerely,

Hirofumi Noguchi, MD, PhD

Department of Regenerative Medicine, Graduate School of Medicine

University of the Ryukyus, 207 Uehara, Nishihara

Okinawa 903-0215, Japan

Tel: +81-98-895-1696; Fax: +81-98-895-1684

This manuscript is a resubmission of an earlier submission. The following is a list of the peer review reports and author responses from that submission.

Round  1

Reviewer 1 Report

The manuscript by Nakaschima and co-authors describes the differential proteomic analysis in adipose stem cells isolated from C57BL/6 and SCID mice by LC-MS/MS in order to address the immunomodulatory potential of these mesenchymal stem cells.

The issue is very important, however, in this current form the manuscript needs to be strongly improved.

My opinion is that in the current form the manuscript is not suitable for publication. I encourage the resubmission after a strong revision.

Major points:

-The introduction must be significantly revised. The aim of the work must be better emphasized.

-The results are interesting, however, they must be re-organized. In particular, the comparison between the ASCs from C57BL/6 and SCID mice must be clearly and easier presented. This includes the figures.

-The discussion must be rewritten. The manuscript lacks clear conclusions.

Reviewer 2 Report

In this manuscript, Nakashima et al compared the protein expression profile of adipose tissue mesenchymal stem cells they isolated from immunocompetent and immunodeficient mice. With an unclear rationale, the authors concluded that A-MSC from both mice constantly express proteins linked to immune response control at least in vitro. 

This descriptive and comparative study looks very preliminary and its rationale should be more elaborated to help the readers easily following the results. Other experiments are needed to strengthen the quality of the study like the use of additional male samples, the comparison of specific features of the cells, and their immunomodulatory properties. I also strongly advice the authors top rewrite the abstract, the results and the discussion sections.

- How many mice were used in that study?

 Additional major comments:

- Many information on the MSC used are lacking [Passage number, viability level, proliferation potential (cumulative population doubling), initial recovered yield, …]

- How those MSC behave in an inflammatory environment?

- To support the claimed conclusion related to the effect of in vitro culture on the A-MSC phenotype, the authors should compare the plated A-MSC to non-plated ones.

- The main proteins significantly expressed should be reanalyzed by another experimental approach  

- The secretome protein levels should also be checked

Detailed comments:

- The reference list should be revised and more organized. Ex: 3 references for citing dental pulp MSC is too much as the statement is very general, Bosma et al should be inserted, …

- Figure 1 should be completely reorganized for a better clarity. A, should be for photos, B, for FACS analysis and C for differentiation of both groups

- Page 4: Why the amount of recovered protein yield of SCID is 2 times higher (6506 µg/ml) than in C57BL/6 (3105 µg/ml)?

- The quality of the photos related to adipogenic differentiation is not convincing. 

- Please use osteoblasts or Osteocytes!

- The paragraphs and subparagraphs of the results sections should be reorganized

- Paragraphs 2.10 & 2.11: no results description was provided!

- The message of the title should be improved and more attractive

- Several typo and grammatical errors should be corrected